# Dendritic branch structure compartmentalizes voltage-dependent calcium influx in cortical layer 2/3 pyramidal cells

Andrew T Landau[1], Pojeong Park[2†], J David Wong-Campos[2†], He Tian[2], Adam E Cohen[2,3], Bernardo L Sabatini[1*]

[1]Howard Hughes Medical Institute, Department of Neurobiology, Harvard Medical School, Boston, United States; [2]Department of Chemistry and Chemical Biology, Harvard University, Cambridge, United States; [3]Department of Physics, Harvard University, Cambridge, United States

**\*For correspondence:**
bsabatini@hms.harvard.edu

[†]These authors contributed equally to this work

**Abstract** Back-propagating action potentials (bAPs) regulate synaptic plasticity by evoking voltage-dependent calcium influx throughout dendrites. Attenuation of bAP amplitude in distal dendritic compartments alters plasticity in a location-specific manner by reducing bAP-dependent calcium influx. However, it is not known if neurons exhibit branch-specific variability in bAP-dependent calcium signals, independent of distance-dependent attenuation. Here, we reveal that bAPs fail to evoke calcium influx through voltage-gated calcium channels (VGCCs) in a specific population of dendritic branches in mouse cortical layer 2/3 pyramidal cells, despite evoking substantial VGCC-mediated calcium influx in sister branches. These branches contain VGCCs and successfully propagate bAPs in the absence of synaptic input; nevertheless, they fail to exhibit bAP-evoked calcium influx due to a branch-specific reduction in bAP amplitude. We demonstrate that these branches have more elaborate branch structure compared to sister branches, which causes a local reduction in electrotonic impedance and bAP amplitude. Finally, we show that bAPs still amplify synaptically-mediated calcium influx in these branches because of differences in the voltage-dependence and kinetics of VGCCs and NMDA-type glutamate receptors. Branch-specific compartmentalization of bAP-dependent calcium signals may provide a mechanism for neurons to diversify synaptic tuning across the dendritic tree.

## Editor's evaluation

Synaptic changes in forebrain neurons typically require the conjunction of dendritic action potentials and synaptic activation. Together these signals cause nonlinear changes in calcium influx that then drive plasticity. The strength of these interactions can vary in complex ways. This study uses state of the art methods to convincingly show how some of these complexities arise from known properties of neuronal dendrites and synaptic NMDA receptors.

## Introduction

The synaptic tuning properties of neurons are regulated by calcium-dependent plasticity signals, including voltage-dependent calcium influx evoked by back-propagating action potentials (bAPs) (*Magee and Johnston, 1997*; *Sjöström and Nelson, 2002*; *Häusser and Mel, 2003*). Many investigations have focused on timing-dependent plasticity rules, such as spike-timing dependent plasticity

(STDP), in which the delay between synaptic input and the bAP determines if an induction protocol results in long-term potentiation (LTP), depression (LTD), or maintenance of synapse strength (*Feldman, 2012*). However, other studies have shown that the outcome of plasticity induction protocols also depends on the synaptic location within the dendritic tree (*Golding et al., 2002*; *Froemke et al., 2005*; *Sjöström and Häusser, 2006*; *Gordon et al., 2006*). The same protocol can evoke a different sign, magnitude, or temporal profile of synaptic plasticity if applied to synapses in proximal or distal dendritic compartments. Such location-dependent plasticity rules may bolster the integrative capacity of individual neurons by supporting divergent tuning properties in basal and apical dendritic arbors (*Larkum et al., 1999*; *Xu et al., 2012*; *Iacaruso et al., 2017*).

Location-dependent plasticity rules vary along the proximodistal axis of dendrites because bAPs are attenuated as they propagate away from the soma (*Regehr et al., 1989*; *Spruston et al., 1995*; *Golding et al., 2002*; *Waters et al., 2003*; *Froemke et al., 2005*; *Sjöström and Häusser, 2006*). In distal dendritic compartments, smaller amplitude bAPs evoke less voltage-dependent calcium-influx, such that induction protocols that usually evoke LTP result in LTD (*Nevian and Sakmann, 2006*; *Sjöström and Häusser, 2006*). Experimentally boosting distal bAP amplitude can restore LTP by amplifying bAP-dependent calcium influx (*Sjöström and Häusser, 2006*), suggesting that location-dependent variation in synaptic plasticity rules is primarily determined by the local amplitude and timing of bAP-dependent calcium influx.

In principle, bAP-dependent calcium influx can also vary in a branch-specific manner, independent of distance from the soma. Dendritic bAP-dependent calcium influx could be shaped by local variability in ion channel expression, synaptic inputs, or electrotonic impedance due to dendritic branch structure (*Rall and Rinzel, 1973*; *Vetter et al., 2001*; *Frick et al., 2004*; *Losonczy et al., 2008*; *Yaeger et al., 2019*). Branch-specific variation in bAP-dependent calcium influx provides a potential mechanism for diversifying synaptic tuning across the dendritic tree, which is a core feature of computational models of neurons as hierarchical nonlinear integrators (*Poirazi et al., 2003*; *Häusser and Mel, 2003*; *Francioni and Harnett, 2021*; *Bicknell and Häusser, 2021*). The existence of branch-specific variation in bAP-dependent calcium influx would also affect the interpretation of dendritic calcium transients imaged in vivo, which can be evoked by local synaptic integration or global bAP-dependent influx (*Wilson et al., 2016*; *Iacaruso et al., 2017*; *Yaeger et al., 2019*; *Beaulieu-Laroche et al., 2019*; *Kerlin et al., 2019*; *Francioni et al., 2019*).

To discover potential dendrite branch-specific and distance-independent variation in bAP amplitude and bAP-dependent calcium influx, we measured bAP-dependent calcium influx in multiple dendrites of individual cortical L2/3 pyramidal cells in mice. We used somatic and dendritic electrical recordings, two-photon calcium imaging and glutamate uncaging, and dendritic voltage-imaging to experimentally identify dendrite branch-specific changes in bAP amplitude and its consequences for bAP-dependent calcium influx. In some branches, bAPs fail to evoke calcium influx through voltage-gated calcium channels (VGCCs), but still amplify synaptically-mediated calcium influx through NMDA receptors (NMDARs). We demonstrated that these branches successfully propagate bAPs in the absence of synaptic input and contain VGCCs, but exhibit more elaborate dendritic branching, which decreases their electrotonic impedance and reduces the amplitude of the bAP. Our results reveal branch-specific variation in bAP-dependent calcium signals, providing an additional mechanism by which calcium-dependent plasticity induction may vary throughout individual neurons.

## Results

We measured calcium influx in the apical dendrites of cortical L2/3 pyramidal cells to investigate whether voltage-dependent calcium signals are regulated in a branch-specific manner. We acquired whole-cell current-clamp recordings from individual cells filled through the recording pipette with 10 µM Alexa Fluor 594 and 300 µM Fluo-5F to visualize neuronal morphology and monitor changes in intracellular calcium concentration using red and green fluorescence, respectively (*Carter and Sabatini, 2004*). Calcium-dependent fluorescence transients were measured as the change in green relative to red fluorescence ($\Delta G/R$), which is linearly proportional to $\Delta Ca$ and comparable across recordings (*Carter and Sabatini, 2004*). We found that bAP-evoked calcium influx ($\Delta Ca_{AP}$) varies across dendritic spines from within the same cells, even when matched for distance from the soma (*Figure 1*). Across many neurons and recording sites, we observed that $\Delta Ca_{AP}$ was highly correlated in nearby spines ( < 6 µm) and between spines and their parent dendritic shafts (*Figure 1—figure supplement*

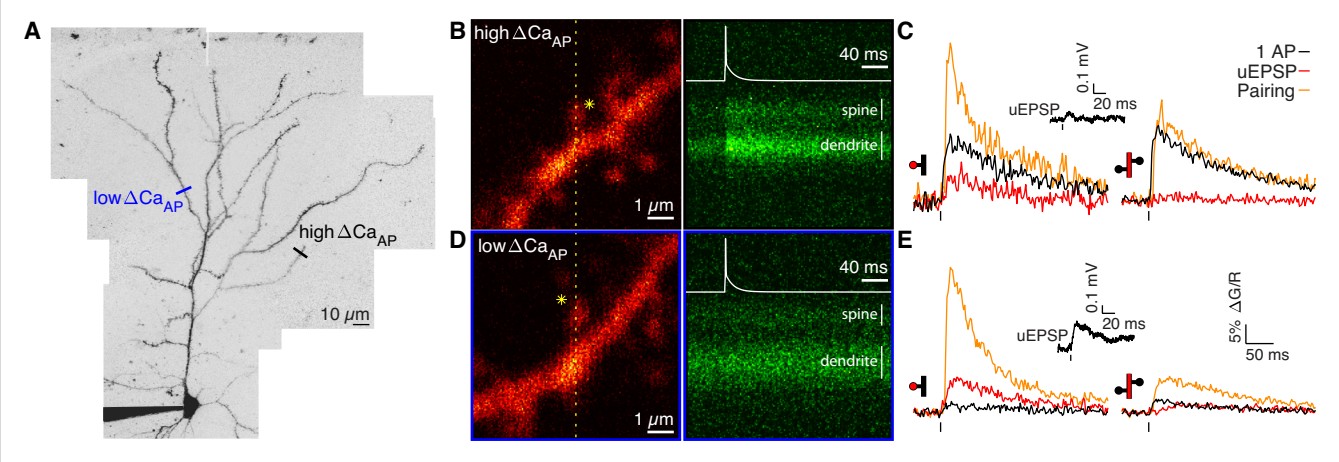

**Figure 1.** Voltage-dependent calcium influx varies in a branch-specific manner throughout L2/3 pyramidal cells. (**A**) Maximum z-projection of Alexa 594 fluorescence showing the full apical dendritic morphology of a cortical L2/3 pyramidal cell. Two sites corresponding to the dendritic regions with low (blue) and high (black) $\Delta Ca_{AP}$ are indicated. (**B**) Frame scan (left) showing high $\Delta Ca_{AP}$ branch. Dotted yellow line indicates the orientation of the line scan used to acquire the data in the kymograph (right) of Fluo-5F fluorescence in the spine and neighboring dendrite evoked by a bAP. Star indicates glutamate uncaging location. White vertical lines indicate ROIs for spine and dendrite. Inset: bAP waveform recorded at the soma. (**C**) Average calcium-dependent fluorescence transients in a high $\Delta Ca_{AP}$ dendritic spine (left) and parent dendrite (right) evoked by 1 bAP, an uEPSP, and pairing of an uEPSP with 1 bAP (bAP evoked 5ms after the uEPSP). Inset: average somatic whole-cell recording of the uEPSPs. In this and all presentations of imaging data, the red shaded area of the spine and dendritic schematic indicates the region from which fluorescence was measured. (**D**) As in B for the low $\Delta Ca_{AP}$ branch. (**E**) As in C for the low $\Delta Ca_{AP}$ branch. Note the difference in $\Delta Ca_{AP}$ (black) between the spines in B-C and D-E. bAP-evoked calcium influx and bAP-dependent amplification are decoupled in L2/3 pyramidal cells.

The online version of this article includes the following figure supplement(s) for figure 1:

**Figure supplement 1.** bAP-evoked calcium influx is regulated across dendritic branches and consistent in neighboring dendritic spines.

*1*), indicating that variations in $\Delta Ca_{AP}$ are regulated across dendritic branches, rather than across individual dendritic spines. These data suggest that voltage-gated calcium channels (VGCCs), which mediate bAP-evoked calcium influx and are responsible for various forms of plasticity (*Kapur et al., 1998*; *Dolmetsch et al., 2001*; *Yasuda et al., 2003*; *Nevian and Sakmann, 2006*; *Scheuss et al., 2006*, *Bloodgood and Sabatini, 2007*; *Brigidi et al., 2019*), can be differentially activated by bAPs in a branch-specific manner within individual L2/3 pyramidal cells.

To investigate the mechanisms of branch-specific variation in voltage-dependent calcium signals, we measured calcium influx in response to glutamate uncaging-evoked excitatory postsynaptic potentials (uEPSPs) as a proxy for synaptically evoked calcium influx ($\Delta Ca_{uEPSP}$). In addition, we measured bAP-dependent amplification of synaptically-evoked calcium influx due to transient relief of the $Mg^{2+}$ block of NMDARs using uEPSP-bAP pairings ($\Delta Ca_{pairing}$). In all uEPSP-bAP pairings, we delayed the bAP by 5ms relative to the glutamate uncaging laser pulse to maximize bAP-dependent amplification of postsynaptic calcium influx and mimic protocols that robustly induce NMDAR-dependent LTP (*Froemke et al., 2005*; *Nevian and Sakmann, 2006*; *Holbro et al., 2010*). Spines on branches with and without bAP-evoked calcium influx showed $\Delta Ca_{uEPSP}$ and supralinear bAP-dependent amplification in $\Delta Ca_{pairing}$ (*Figure 1B–E*; *Holbro et al., 2010*), even though one of the spines exhibited almost no bAP-evoked calcium influx when the bAP was evoked by itself (*Figure 1D–E*). These data suggest that bAP-evoked calcium influx through VGCCs and bAP-dependent amplification of synaptically evoked calcium influx through NMDARs can be regulated independently in a branch-specific manner (*Bloodgood and Sabatini, 2007*; *Holbro et al., 2010*).

## bAP-evoked calcium influx and bAP-dependent amplification are decoupled in L2/3 pyramidal cells

To systematically investigate the regulation of bAP-evoked calcium influx and bAP-dependent amplification of synaptically evoked calcium influx, we recorded $\Delta Ca_{AP}$, $\Delta Ca_{uEPSP}$, and $\Delta Ca_{pairing}$ in 97 dendritic spines from 33 cells and 15 mice (N = 97/33/15) spanning the full proximodistal extent of the dendrites. To focus our analysis on the supralinear component of $\Delta Ca_{pairing}$, which is primarily carried by NMDARs

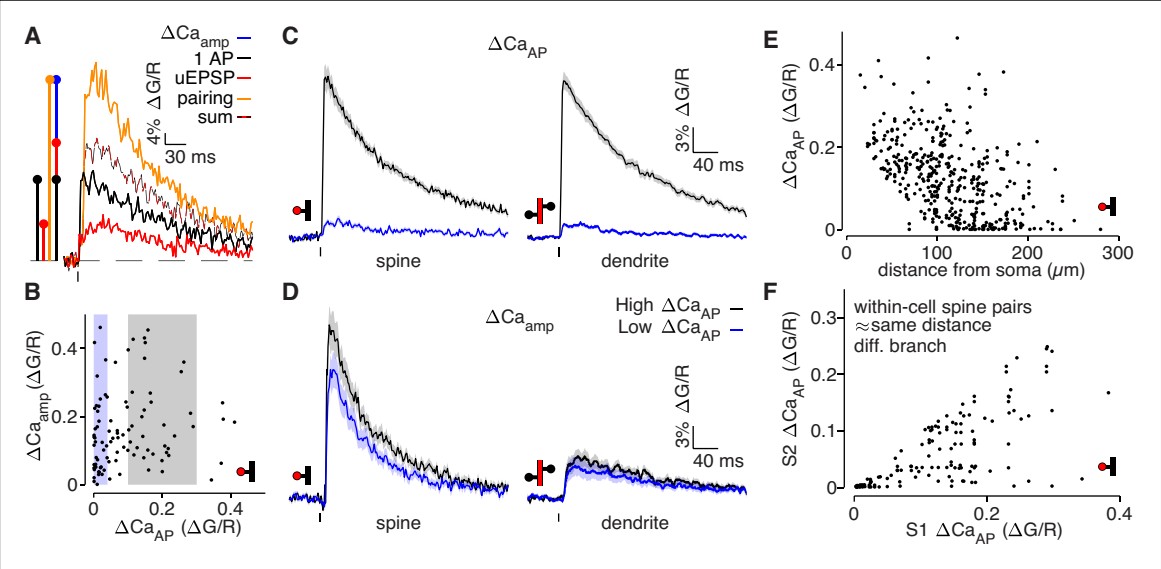

**Figure 2.** bAP-evoked calcium influx and bAP-dependent amplification are decoupled in L2/3 pyramidal cells. (**A**) Average calcium-dependent fluorescence transients recorded from a dendritic spine and a schematic of the method for computing the amplification of calcium influx caused by interaction of the bAP and uEPSP, which results in additional influx (blue) relative to the sum of the bAP and uEPSP components measured independently (black/red). (**B**) bAP-dependent amplification vs. bAP-evoked calcium influx. Shaded areas indicate selection of high $\Delta Ca_{AP}$ (gray) and low $\Delta Ca_{AP}$ (blue) spines used in panels C and D. (**C**) $\Delta Ca_{AP}$ in high (black) and low (blue) $\Delta Ca_{AP}$ dendritic spines (left) and parent dendritic shafts (right). In this and all presentations of average fluorescence transients, the line and shaded areas show the mean and standard error around the mean. (**D**) $\Delta Ca_{amp}$ in high (black) and low (blue) $\Delta Ca_{AP}$ dendritic spines (left) and parent dendritic shafts (right). (**E**) $\Delta Ca_{AP}$ across distance from soma in all dendritic spines (N = 526/95/29). (**F**) Comparison of $\Delta Ca_{AP}$ between pairs of dendritic spines (i.e. S1 & S2) that were recorded from the same cell, on different dendritic branches, and at a similar distance from the soma ( < 20 μm relative distance from soma). Each pair was sorted so the abscissa indicates the $\Delta Ca_{AP}$ of the spine with higher calcium influx (N = 94/22/6).

The online version of this article includes the following figure supplement(s) for figure 2:

**Figure supplement 1.** bAP-dependent amplification is not determined by glutamate uncaging properties.

(**Holbro et al., 2010**), we measured amplification as $\Delta Ca_{amp} = \Delta Ca_{pairing} - (\Delta Ca_{AP} + \Delta Ca_{uEPSP})$ (**Figure 2A**). We found that $\Delta Ca_{AP}$ and $\Delta Ca_{amp}$ were uncorrelated across dendritic spines in L2/3 pyramidal cells (**Figure 2B**). To visualize the divergence between $\Delta Ca_{AP}$ and $\Delta Ca_{amp}$, we grouped the data into high $\Delta Ca_{AP}$ ($0.1 < \Delta Ca_{AP} < 0.3$, selected to avoid outliers), and low $\Delta Ca_{AP}$ ($\Delta Ca_{AP} < 0.04$) populations (**Figure 2B**) based on the amplitude of bAP-evoked calcium influx (**Figure 2C–D**). We plot data with this selection criterion throughout the figures to highlight differences across branches but focus our statistical analyses on the full distribution of data which is presented as scatter plots in each figure. The same criterion is used throughout all figures except in Figures 5B–D. Low $\Delta Ca_{AP}$ branches exhibited almost no calcium influx evoked by bAPs but still exhibited large bAP-dependent amplification (**Figure 2C–D**). $\Delta Ca_{AP}$ and $\Delta Ca_{amp}$ showed similar patterns in dendritic spines and parent dendritic shafts (**Figure 2C–D**). We found that although $\Delta Ca_{AP}$ attenuates with distance from the soma (**Figure 2E** and **Figure 2—figure supplement 1B**), it exhibits significant variability in distal branches, such that pairs of dendritic spines that are on different branches within the same neuron but at a similar distance from the soma sometimes had large differences in calcium influx (**Figure 2E–F**). $\Delta Ca_{amp}$ varied over a similar range at all distances from the soma (**Figure 2—figure supplement 1A**). The divergence between $\Delta Ca_{AP}$ and $\Delta Ca_{amp}$ was not related to the degree of NMDAR activation (**Figure 2—figure supplement 1C**) nor the uEPSP amplitude or time-course (**Figure 2—figure supplement 1D-H**). Furthermore, we observed low $\Delta Ca_{AP}$ branches in the presence of MNI-Glutamate, which is an antagonist of ionotropic GABA receptors, indicating that low $\Delta Ca_{AP}$ branches are not caused by inhibition (**Figure 2B**). We confirmed this with the selective GABA-A receptor antagonist 10 μM SR 95531 (data not shown). Together, these data indicate that $\Delta Ca_{AP}$ and $\Delta Ca_{amp}$ are regulated independently in the dendrites of L2/3 pyramidal cells.

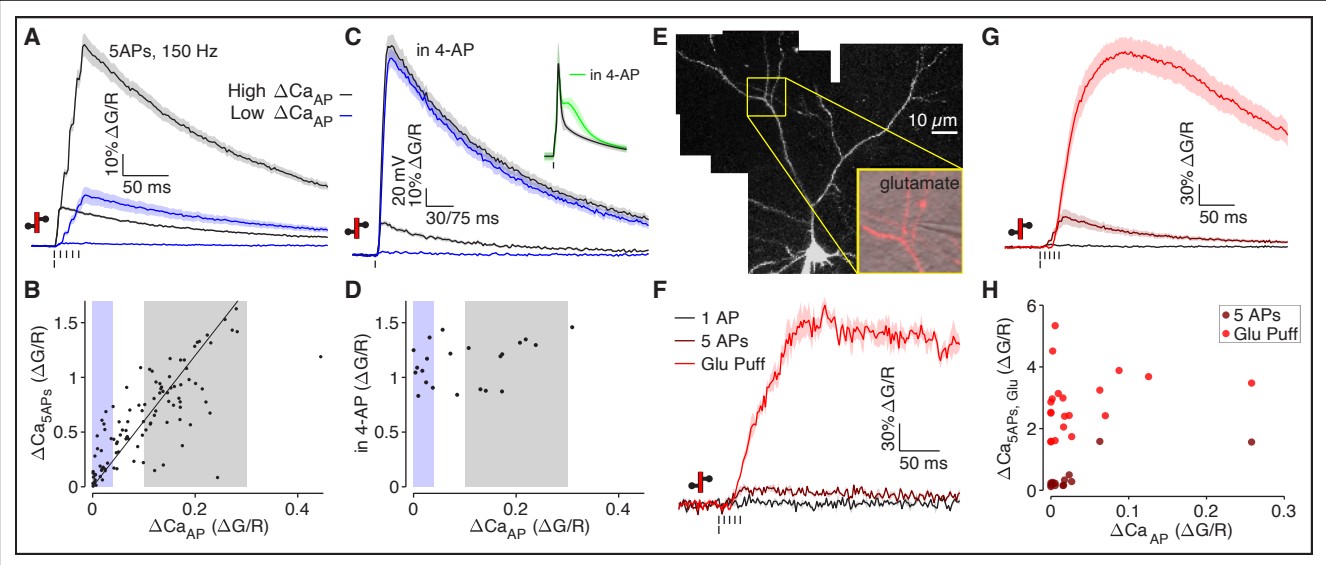

**Figure 3.** Low $\Delta Ca_{AP}$ dendrites contain voltage-gated calcium channels. (**A**) Calcium-dependent fluorescence transients evoked by 1 bAP or 5 bAPs at 150 Hz in high (black) and low (blue) $\Delta Ca_{AP}$ dendrites. (**B**) Comparison of peak calcium influx evoked by 1 bAP or 5 bAPs in each dendrite. Gray and blue patches indicate selection of high (gray) and low (blue) $\Delta Ca_{AP}$ spines used in panel A. The line y = 6 x is plotted for reference. N = 109/33/9, $R^2$ = 0.609, $F$ = 166, p = 1.5 × $10^{-23}$.(**C**) Calcium-dependent fluorescence transients evoked by 1 bAP before and after application of 2 mM 4-AP in high (black) and low (blue) $\Delta Ca_{AP}$ branches. Inset: average AP waveform before and after application of 4-AP. (**D**) Comparison of peak calcium influx evoked by 1 bAP before and after application of 2 mM 4-AP. Gray and blue patches indicate selection of high (gray) and low (blue) $\Delta Ca_{AP}$ spines used in panel C. There was no significant correlation: N = 22/7/3, $R^2$ = 0.135, $F$ = 3.11, p = 0.093. (**E**) Maximum z-projection of Alexa 594 fluorescence from a L2/3 pyramidal cell with differential-interference contrast image overlaid showing glutamate puffing pipette. The full apical dendritic morphology was not imaged. (**F**) Calcium-dependent fluorescence transients evoked by 1 bAP, 5 bAPs at 150 Hz, and a 1 mM glutamate puff in the example branch from panel E. (**G**) Average calcium-dependent fluorescence transients evoked by 1 bAP, 5 bAPs at 150 Hz, and a 1 mM glutamate puff across all branches recorded. (**H**) Comparison of peak calcium influx evoked by bAP, 5 bAPs at 150 Hz, and a 1 mM glutamate puff in all branches. The response to 5 bAPs was not recorded in all dendrites (N = 20/6/3).

The online version of this article includes the following figure supplement(s) for figure 3:

**Figure supplement 1.** Glutamate puff-evoked calcium signals are mediated by voltage-gated calcium channels.

We considered three distinct mechanistic explanations for why low $\Delta Ca_{AP}$ branches exhibit bAP-dependent amplification ($\Delta Ca_{amp}$) but fail to exhibit bAP-evoked calcium influx ($\Delta Ca_{AP}$):

1. Low $\Delta Ca_{AP}$ branches contain fewer (or no) voltage-gated calcium channels (VGCCs).
2. bAPs fail to backpropagate into low $\Delta Ca_{AP}$ branches without the support of EPSPs.
3. bAP amplitude in low $\Delta Ca_{AP}$ branches does not exceed the threshold for VGCC opening.

Although each of these mechanisms could be realized in several ways, they represent general explanations that we tested directly.

## Low $\Delta Ca_{AP}$ branches contain voltage-gated calcium channels

Low $\Delta Ca_{AP}$ branches may contain functional VGCCs that are not opened by a single bAP but do open in response to stronger depolarizations. To examine this possibility, we imaged calcium influx evoked by a burst of 5 bAPs at 150 Hz in dendritic shafts located >75 μm from the soma in L2/3 pyramidal cells (*Figure 3A*; *Larkum et al., 2007*). In most low $\Delta Ca_{AP}$ branches, calcium influx could be evoked by a burst of 5 bAPs (*Figure 3A–B*). However, the calcium influx evoked by a burst of 5 bAPs in high $\Delta Ca_{AP}$ branches was higher than in low $\Delta Ca_{AP}$ branches (*Figure 3A–B*). These data are consistent with the presence of VGCCs in low $\Delta Ca_{AP}$ branches, although they do not distinguish whether there is a lower density of VGCCs (or a higher voltage threshold), or if bursts of bAPs are less effective at depolarizing low $\Delta Ca_{AP}$ branches.

We attempted to make the depolarization evoked across the dendritic tree more uniform by triggering bAPs in the presence of the potassium channel antagonist 4-AP (*Figure 3C–D*; *Hoffman et al., 1997*). In all somatic recordings, application of 2 mM 4-AP increased the after-depolarization of the bAP (*Figure 3C*, inset), confirming the efficacy of the drug. Applying 4-AP significantly increased $\Delta Ca_{AP}$ in every branch, regardless of how much calcium influx was evoked in control conditions (*Figure 3C–D*). Because $\Delta Ca_{AP}$ evoked in the presence of 4-AP is similar in high and low $\Delta Ca_{AP}$ branches (*Figure 3C*), we conclude that VGCCs are present in all dendritic compartments and mediate substantial calcium influx in response to sufficiently large depolarizations.

To determine if VGCCs can be effectively opened by local depolarization in low $\Delta Ca_{AP}$ branches, we measured calcium influx evoked by direct application of 1 mM glutamate delivered via a patch pipette using a picospritzer (*Figure 3E–H*). We added 20 µM CPP and 50 µM MK-801 to the puff and bath solutions to block NMDARs and limit our measurements to calcium influx mediated by VGCCs. In every dendrite examined, direct application of glutamate led to significant calcium influx, regardless of $\Delta Ca_{AP}$ (*Figure 3F–H*). The calcium signal was not an artifact of mechanical displacement due to the picospritzer (*Figure 3—figure supplement 1A-B*) and did not include an NMDAR component (*Figure 3—figure supplement 1C-D*), consistent with calcium influx mediated by VGCCs. We note that the large calcium signals evoked by glutamate demonstrate that the similarity in bAP-evoked calcium influx between high and low $\Delta Ca_{AP}$ measured in 4-AP was not due to saturation of the calcium indicator. Collectively, our data indicate that all dendritic branches contain VGCCs, which

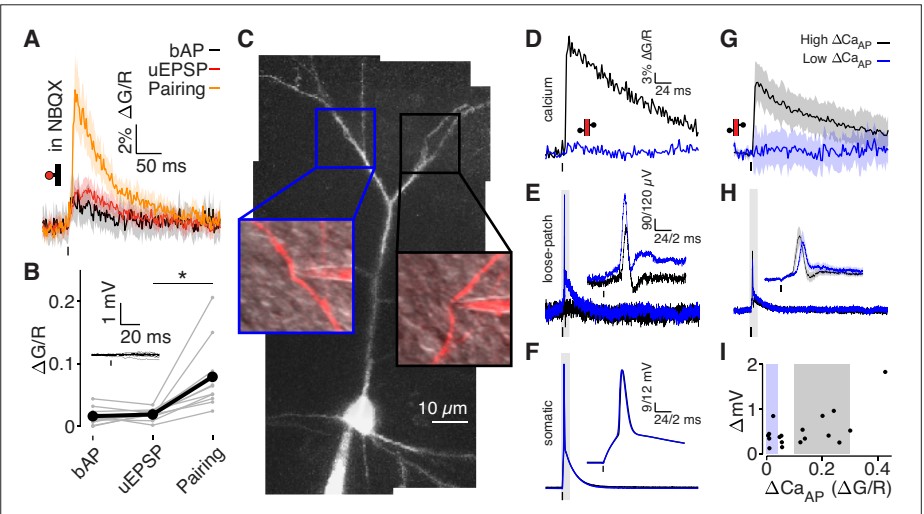

**Figure 4.** bAPs propagate to low $\Delta Ca_{AP}$ dendrites in the absence of EPSPs. (**A**) Calcium-dependent fluorescence transients evoked by bAPs, uEPSPs, and bAP/uEPSP pairings in the presence of 10 µM NBQX. (**B**) Comparison of peak calcium influx in spines evoked by bAP, uEPSP, and bAP/uEPSP pairing, in the presence of 10 µM NBQX. Inset: whole-cell recordings from glutamate uncaging in the presence of NBQX. Calcium influx evoked by pairing was significantly higher than by the uEPSP alone, *t*-test: N = 12/8/2, t = 4.01, p = 5.8 × 10⁻⁴. (**C**) Maximum z-projection of Alexa 594 fluorescence from a L2/3 pyramidal cell with a whole-cell recording in the soma and consecutive loose-patch recordings in two branches. Insets: red fluorescence overlaid on differential interference contrast image of loose-patch configuration for each dendrite. The full apical dendritic morphology was not imaged. (**D**) Calcium-dependent fluorescence transients evoked by bAPs in each dendrite from panel C, recorded during loose-patch recordings. (**E**) Electrical signals evoked by bAPs measured with dendritic loose-patch recordings in each dendrite from panel C. Inset: expanded trace from gray shaded region. (**F**) Somatic whole-cell recording of bAPs during each loose-patch recording from panel C. Inset: expanded trace from gray shaded region. (**G**) Average calcium-dependent fluorescence transients evoked by bAPs in high (black) and low (blue) $\Delta Ca_{AP}$ branches during loose-patch recordings. (**H**) Average electrical signal evoked by bAPs in high (black) and low (blue) $\Delta Ca_{AP}$ branches measured with loose-patch recordings. (**I**) Comparison of peak calcium influx and peak electrical signal evoked in each dendrite by bAPs. Patches indicate selection of high (gray) and low (blue) $\Delta Ca_{AP}$ branches used in panels G and H (N = 19/15/9).

The online version of this article includes the following figure supplement(s) for figure 4:

**Figure supplement 1.** Loose-patch recording properties.

can be opened by local depolarization. However, a specific population of low $\Delta Ca_{AP}$ branches exhibit little to no calcium influx in response to back-propagating bAPs.

## bAPs propagate to low $\Delta Ca_{AP}$ branches in the absence of EPSPs

We only observe significant bAP-dependent calcium influx in low $\Delta Ca_{AP}$ branches if the bAP is paired with synaptic input evoked by glutamate uncaging. Despite the small size of the uEPSP, it is possible that bAPs only back-propagate to low $\Delta Ca_{AP}$ branches when assisted by local depolarization (*Magee and Johnston, 1997*; *Gasparini et al., 2007*). Since most of the depolarization evoked by EPSPs is carried by AMPARs, we measured $\Delta Ca_{AP}$ , $\Delta Ca_{uEPSP}$ , and $\Delta Ca_{pairing}$ after applying the selective AMPAR antagonist, NBQX (*Figure 4A–B*). We observed bAP-dependent amplification of the remaining NMDAR-mediated calcium influx in all dendritic spines examined, despite fully blocking the depolarization evoked by glutamate uncaging (*Figure 4A–B*). As expected, glutamate uncaging evokes some calcium influx via NMDARs in the presence of NBQX, since NMDARs are only partially blocked by Mg$^{2+}$ at resting potentials (*Jahr and Stevens, 1990*) and the driving force on calcium is extremely high. These data suggest that bAPs propagate to low $\Delta Ca_{AP}$ branches without an uEPSP-mediated local depolarization.

To test whether bAPs propagate to low $\Delta Ca_{AP}$ branches in the absence of glutamate receptor activation, we acquired loose-patch recordings in current-clamp from 19 dendrites in 15 neurons while simultaneously recording from the soma to measure the dendritic electrical signal evoked by a bAP (*Figure 4C–I*). Simultaneous two-photon imaging was used to monitor dendritic calcium influx. All recording sites were ≥75 µm from the soma. In the neuron shown in *Figure 4C*, we acquired consecutive loose-patch recordings from one high $\Delta Ca_{AP}$ branch and one low $\Delta Ca_{AP}$ branch. Only the high $\Delta Ca_{AP}$ branch had detectable $\Delta Ca_{AP}$ , but both dendritic recordings exhibited clear electrical signals evoked by the bAP (*Figure 4D–F*). We observed bAP-evoked electrical signals in all dendrites, even though only some exhibited $\Delta Ca_{AP}$ (*Figure 4G–I*). These loose-patch recordings do not permit direct measurement of intracellular bAP amplitude (*Figure 4—figure supplement 1*); however, they demonstrate that bAPs propagate to low $\Delta Ca_{AP}$ branches without the support of EPSPs.

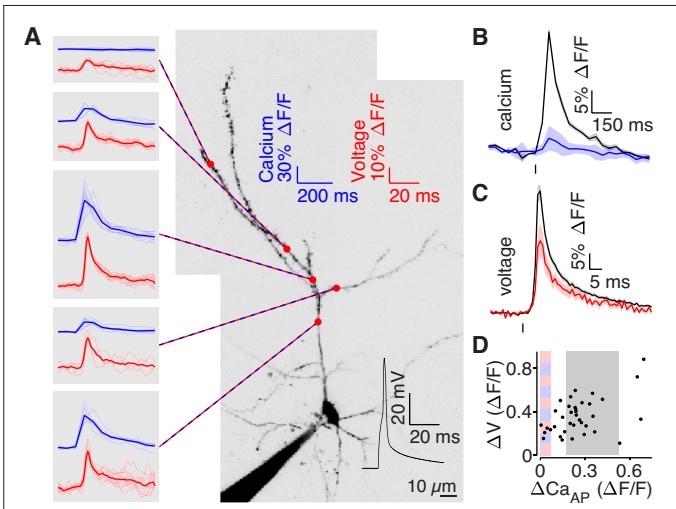

**Figure 5.** bAP amplitude is attenuated in low $\Delta Ca_{AP}$ branches. (**A**) Mean calcium-dependent (blue) and voltage-dependent (red) fluorescence transients evoked by 1 bAP (left) measured in the 5 branches indicated in the maximum z-projection of Alexa 594 fluorescence from a L2/3 pyramidal cell (right).The black trace shows the bAP waveform recorded from the soma during calcium and voltage imaging. Note that calcium- and voltage-dependent transients are shown on ~10 fold different time scales. The full apical dendritic morphology was not imaged. (**B**) Average calcium-dependent fluorescence transient in high (black) and low (blue) Δ Ca$_{AP}$ branches evoked by 1 bAP. (**C**) Average voltage-dependent fluorescence transient in high (black) and low (red) Δ Ca$_{AP}$ branches evoked by 1 bAP. (**D**) Comparison of the peak voltage- and calcium-dependent fluorescence transient evoked by 1 bAP. Shaded regions indicate selection criterion for high (gray) and low (blue/red) $\Delta Ca_{AP}$ branches. Calcium-dependent fluorescence predicted voltage-dependent fluorescence, N = 37, R$^2$ = 0.243, *F* = 11.2, p = 1.9 × 10$^{-3}$ and the y-intercept was significantly above 0, y-int = 0.235, t(35)=5.34, p = 5.6 × 10$^{-3}$.

## bAP amplitude is attenuated in low $\triangle Ca_{AP}$ branches

To measure the intracellular dendritic voltage evoked by bAPs, we performed interleaved dendritic voltage and calcium imaging while performing somatic whole-cell current-clamp recordings (*Figure 5*). Voltage imaging was achieved by expressing a highly-sensitive near-infrared genetically-encoded voltage indicator, QuasAr6a (*Wang et al., 2021*). Both voltage and calcium signals were recorded via structured illumination one-photon microscopy. All recording sites were ≥75 μm from the soma (except for the closest site in panel A). Consistent with our results from *Figure 4*, we observed bAP-dependent voltage signals in all branches, including those that did not exhibit bAP-evoked calcium influx (*Figure 5*). To visualize differences in high and low $\triangle Ca_{AP}$ branches, we adjusted our selection criterion to account for differences in the range of $\triangle Ca_{AP}$ measured with two-photon and one-photon calcium imaging (the selection range described above was scaled by 1.75, *Figure 5B–D*). Although low $\triangle Ca_{AP}$ branches exhibited clear bAP-evoked voltage waveforms (*Figure 5C*), the amplitudes of the dendritic bAP-evoked voltage waveforms were attenuated in low $\triangle Ca_{AP}$ branches (*Figure 5C–D*). The full-width half-max of the voltage waveforms was 6.2 ±0.9ms for high $\triangle Ca_{AP}$ branches and 8.0 ±4.1ms for low $\triangle Ca_{AP}$ branches. These data demonstrate that bAP amplitude is reduced in low $\triangle Ca_{AP}$ branches.

Collectively, our data support the conclusion that low $\triangle Ca_{AP}$ branches exhibit less bAP-evoked calcium influx because of branch-specific reductions in local bAP amplitude. First, our experiments with bursts of 5 bAPs, application of 4-AP, and glutamate application show that low $\triangle Ca_{AP}$ branches contain VGCCs that open in response to large depolarizations (*Figure 3*). Second, our experiments with NBQX and dendritic loose-patch recordings show that bAPs successfully propagate to low $\triangle Ca_{AP}$ branches in the absence of synaptic input (*Figure 4*). Third, our voltage imaging experiments show that the amplitude of bAPs is attenuated in low $\triangle Ca_{AP}$ branches (*Figure 5*). These data are consistent with the mechanistic explanation that bAPs fail to evoke calcium influx in low $\triangle Ca_{AP}$ branches because the local bAP amplitude does not exceed the threshold for VGCC opening. Therefore, we investigated why bAPs are selectively attenuated in low $\triangle Ca_{AP}$ branches.

## bAP-evoked calcium influx attenuates at branch points

bAP amplitude and bAP-evoked calcium influx can attenuate at branch points due to dendritic impedance mismatch and localized expression of potassium channels (*Spruston et al., 1995*; *Williams and Stuart, 2000*; *Vetter et al., 2001*; *Frick et al., 2003*; *Cai et al., 2004*; *Gasparini et al., 2007*; *Harnett et al., 2013*). Consistent with these observations, we found that $\triangle Ca_{AP}$ often attenuates across branch points, as revealed by comparing differences in $\triangle Ca_{AP}$ at pairs of sites within the same dendritic segment or across one dendritic branch point (*Figure 6—figure supplement 1A-C*). The ratio of $\triangle Ca_{AP}$ (influx at distal site divided by proximal site) tends to be smaller for pairs of sites spanning a branch point than for those within the same segment (rank-sum: p = *0.0145*, *Figure 6—figure supplement 1C*). This result was corroborated by the observation that low $\triangle Ca_{AP}$ branches typically had higher dendritic branch order (defined as the number of on-path branch points between the soma and the recording site) than high $\triangle Ca_{AP}$ branches (*Figure 6—figure supplement 1D*). While these data suggest a relationship between $\triangle Ca_{AP}$ and the pattern of dendritic branching, they do not fully account for the variance in $\triangle Ca_{AP}$ observed across branches in L2/3 pyramidal cells (*Figure 6—figure supplement 1D*, $R^2$ = 0.178).

## Low $\triangle Ca_{AP}$ branches have a more elaborate branch structure than high $\triangle Ca_{AP}$ branches

The electrotonic impedance (i.e., passive input resistance) of a dendrite scales inversely with the surface area of membrane. Therefore, if low $\triangle Ca_{AP}$ branches are within sections of the dendritic tree that contain more extensive branching, then they have a smaller impedance, which might reduce bAP amplitude and $\triangle Ca_{AP}$ . To examine this possibility, we developed a metric based on the dendritic morphology called 'branch complexity' that accounts for all nearby dendritic branches and is independent of branch order. This approach measures the total length of dendrite near a recording site with an exponential filter to discount branches that are further from the site (so ithas units of millimeters). We measured branch complexity over a length constant of $\lambda$ = 145 μm because propagating bAPs simultaneously depolarize ~145 μm of dendrite in L2/3 pyramidal cells (conduction velocity is 154.9 μm/ms and AP duration is 0.94ms, *Figure 6—figure supplement 2*).

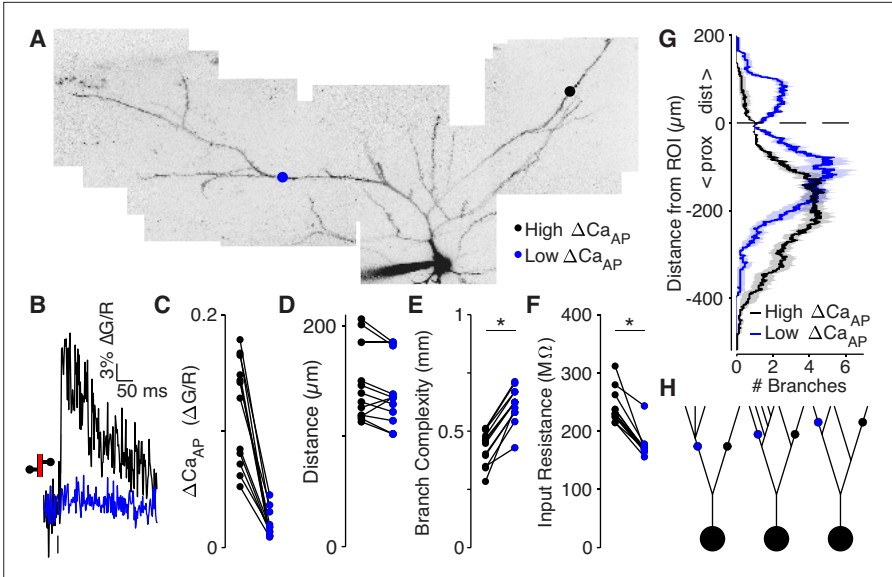

**Figure 6.** Low ΔCa_AP dendrites have a more elaborate branch structure than high ΔCa_AP dendrites. (**A**) Maximum z-projection of Alexa 594 fluorescence from a L2/3 pyramidal cell. Dots indicate location of calcium imaging sites for a high (black) and low (blue) ΔCa_AP branch. The two sites are approximately the same distance from the soma (high ΔCa_AP: 145 µm, low ΔCa_AP: 131 µm). (**B**) Calcium-dependent fluorescence transient evoked by bAP in high (black) and low (blue) ΔCa_AP sites from panel A. (**C**) Comparison of ΔCa_AP in distance-matched, within-cell pairs. N = 20/8/8, N = 12 pairs, Ratio: 0.17, (range: 0.06–0.29). (**D**) Comparison of the distances from the soma for each pair of recording sites. (**E**) Comparison of branch complexity in high (black) and low (blue) ΔCa_AP sites, using a distance-discounted measurement of nearby dendritic branches. Branch complexity was significantly higher in low ΔCa_AP sites, U-Test, U = 215, z = 3.73, p = 1.9 × 10⁻⁴. (**F**) Comparison of input resistance for high (black) and low (blue) ΔCa_AP sites shown in panels C and D, measured in computational simulations of compartment models of each cell in NEURON. Input resistance was significantly lower in low ΔCa_AP sites, U-Test, U = 213, z = 3.61, p = 3.1 × 10⁻⁴. (**G**) Average number of branches at a given distance from a recording site for high (black) and low (blue) ΔCa_AP dendrites. This curve was multiplied by a symmetric exponential filter with a length constant of 145 µm and then integrated to compute the branch complexity in panel E. Mean ± SEM. (**H**) Schematic of morphologies with same branch order but different branch complexity. The blue and black site in each neuron have the same branch order, but the branch complexity of the blue site is higher due to the number of branches distal to the site (left), the number of branches on a sister dendrite (middle), and the distance from a previous branch (right).

The online version of this article includes the following figure supplement(s) for figure 6:

**Figure supplement 1.** bAP-evoked calcium influx attenuates around branch points.

**Figure supplement 2.** Back-propagating APs span 145 µm in L2/3 cell dendrites.

**Figure supplement 3.** Branch complexity is correlated with input resistance.

We compared the branch complexity of pairs of dendritic recording sites within the same cell that were located approximately the same distance from the soma but for which there was a large difference in $\Delta Ca_{AP}$ (*Figure 6A–E*). The low $\Delta Ca_{AP}$ site had a higher branch complexity than the high $\Delta Ca_{AP}$ site in all pairs (*Figure 6E*), indicating that low $\Delta Ca_{AP}$ sites are surrounded by more extensive branching than high $\Delta Ca_{AP}$ sites. To confirm that branch complexity is inversely proportional to input resistance, we reconstructed all cells used in this analysis as biophysical compartment models in the NEURON simulation environment and directly measured the input resistance by injecting small hyperpolarizing currents into each recording site (*Carnevale and Hines, 2006*). In every pair, the low $\Delta Ca_{AP}$ site had a smaller input resistance than the high $\Delta Ca_{AP}$ site (*Figure 6F*). Additionally, branch complexity was inversely correlated with input resistance ($R^2$ = 0.93 for $\lambda$ = 145 µm), confirming its validity as a measure of electrotonic properties (*Figure 6—figure supplement 3A-B*). These results and conclusions hold for measurements of branch complexity over length constants ranging from 5 to 400 µm (*Figure 6—figure supplement 3C*).

To inspect why low $\Delta Ca_{AP}$ branches had a higher branch complexity, we plotted the average number of dendritic branches distal and proximal to the target site (*Figure 6G*). Low $\Delta Ca_{AP}$ branches

are immediately proximal to dendritic branch elaboration (e.g. *Figure 6A* c.f. with *Figure 6H*, left), and closer to sister dendrites emerging proximal to the measurement site (c.f. with *Figure 6H*, middle and right). These results demonstrate that low $\Delta Ca_{AP}$ branches are surrounded by more elaborate branching patterns than high $\Delta Ca_{AP}$ branches, suggesting that dendritic branch structure may be sufficient to selectively reduce the amplitude of bAPs.

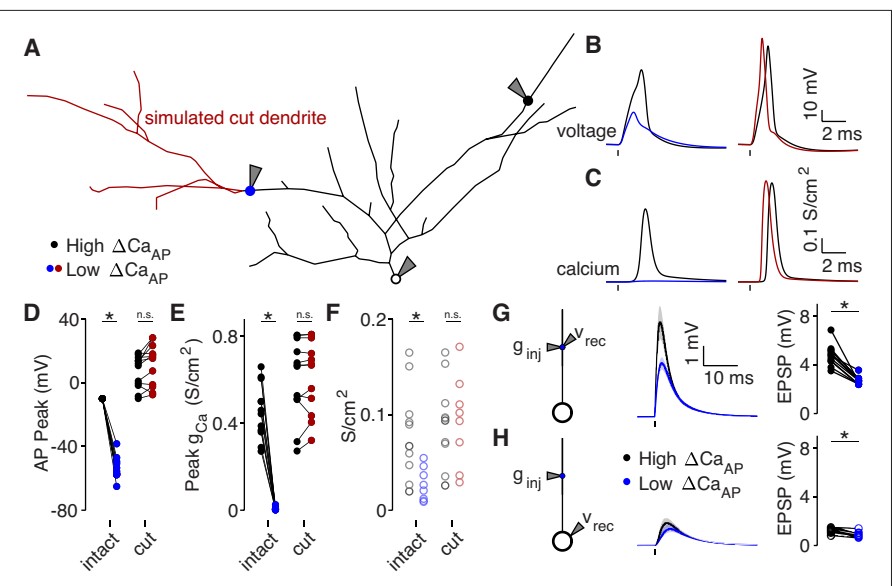

**Figure 7.** Dendritic branch structure is sufficient to explain reductions in bAP-evoked calcium influx. (**A**) Schematic of a neuron morphology used for NEURON simulations derived from the neuron shown in *Figure 6A*. Blue and black dots indicate high (black) and low (blue) $\Delta Ca_{AP}$ recording sites. Red dendrites indicate section of dendrite that was computationally 'cut' in a subset of experiments. Recordings and injections were performed at the sites indicated with the gray triangles. (**B**) Dendritic voltage recordings of bAPs from high (black) and low (blue/red) $\Delta Ca_{AP}$ sites in panel A in NEURON for an intact cell (left) or with cut dendrites (right). (**C**) Calcium conductances recorded from high (black) and low (blue/red) $\Delta Ca_{AP}$ sites in panel A evoked by a back-propagating bAP in NEURON for an intact cell (left) or with cut dendrites (right). (**D**) Comparison of peak bAP voltage for each high/low $\Delta Ca_{AP}$ pair recorded in NEURON from same sites as experimental recordings, for intact and cut cells (N = 12). Dendritic potassium channel density was fit so the high $\Delta Ca_{AP}$ site would have a peak of –10 mV in intact cells. AP Peak was significantly reduced in low $\Delta Ca_{AP}$ sites in the intact cells (U-Test, U = 78, z = 4.13, p = 3.7 × 10$^{-5}$) but not different in cut cells (U-Test, U = 168, z = 1.01, p = 0.31). (**E**) Comparison of peak calcium conductance during bAP for each high/low $\Delta Ca_{AP}$ site pair, as in panel D. Peak calcium conductance was significantly reduced in low $\Delta Ca_{AP}$ sites in the intact cells (U-Test, U = 78, z = 4.13, p = 3.7 × 10$^{-5}$) but not different in cut cells (U-Test, U = 155, z = 0.26, p = 0.79). (**F**) A-Type potassium channel density required for the bAP to peak at –10 mV for each site in intact or cut cells. A-Type potassium channel density was significantly reduced in low $\Delta Ca_{AP}$ sites in the intact cells (U-Test, U = 160, z = 2.58, p = 9.7 × 10$^{-3}$) but not different in cut cells (U-Test, U = 120, z = 0.42, p = 0.67). (**G**) Left: a synaptic conductance was injected into each site and the local EPSP was recorded in NEURON. Middle: dendritic EPSPs recorded in high and low $\Delta Ca_{AP}$ dendritic sites. Right: comparison of dendritic EPSP amplitudes recorded in each high/low $\Delta Ca_{AP}$ pair. Mean ± SEM. Dendritic EPSP amplitude was significantly reduced in low $\Delta Ca_{AP}$ sites (U-Test, U = 79, z = 4.07, p = 4.7 × 10$^{-5}$). (**H**) Same as in G, but with synaptic conductance injected into dendrite and recorded in the soma. Somatic EPSP amplitude was significantly reduced in low $\Delta Ca_{AP}$ sites (U-Test, U = 101, z = 2.8, p = 5.1 × 10$^{-3}$), but had a smaller effect size than difference in dendritic EPSP amplitude (see text).

The online version of this article includes the following figure supplement(s) for figure 7:

**Figure supplement 1.** Schematics of bAP amplitude and peak calcium conductance throughout NEURON compartment models.

**Figure supplement 2.** Branch differences are normalized by blocking A-Type potassium channels or cutting dendrites.

## Dendritic branch structure is sufficient to explain reductions in bAP-evoked calcium influx

To determine if dendritic branch structure is sufficient for reducing $\Delta Ca_{AP}$, we reproduced our experimental results using a biophysical compartment model. We held all physiological parameters constant throughout each model except for the dendritic branch structure, allowing us to specifically measure the effects of dendritic morphology between high and low $\Delta Ca_{AP}$ branches (*Vetter et al., 2001*). Using NEURON compartmental models of 8 reconstructed L2/3 pyramidal cells (all from the pairwise analysis described in *Figure 6*), we evoked a bAP with somatic current injection and measured the intracellular voltage waveform and the resulting calcium conductance at the same sites in which we had imaged calcium influx in acute brain slices (*Figure 7A–E*). For each cell, we fit the density of dendritic A-Type potassium channels such that the bAP peaked at –10 mV in the high $\Delta Ca_{AP}$ site (*Figure 7D*), which is consistent with the amplitude of bAPs measured ~100 μm from the soma in layer 2–3 pyramidal cells with dendritic whole-cell recordings (*Larkum et al., 2007*; *Smith et al., 2013*). We used this density of A-Type potassium channels throughout the entire dendritic tree of each cell. Under these conditions, for every within-cell pair, the low $\Delta Ca_{AP}$ site had smaller bAP amplitude and peak calcium conductance, despite having the same channel densities as the high $\Delta Ca_{AP}$ site (*Figure 7B–E*). As an additional measure of dendritic excitability, we plotted the A-Type potassium channel density required for bAPs to peak at –10 mV in each dendritic site (*Figure 7F*). Low $\Delta Ca_{AP}$ sites required lower potassium channel densities for their bAPs to peak at –10 mV, indicating that the branch structure surrounding low $\Delta Ca_{AP}$ sites is less excitable than for high $\Delta Ca_{AP}$ sites (*Figure 7F*). To visualize the spread of bAP-evoked voltage and calcium signals throughout the dendritic tree, we plotted two example cells with the color of each segment indicating the peak voltage and calcium conductance (*Figure 7—figure supplement 1*). These data demonstrate that dendritic branch structure is sufficient to impact $\Delta Ca_{AP}$ by altering the dendritic bAP amplitude in a branch-specific manner.

We performed a simulated 'cutting' experiment in which we removed dendritic branches from the reconstruction that were closer to the low $\Delta Ca_{AP}$ site than the high $\Delta Ca_{AP}$ site (*Figure 7A*), to determine if simplifying the dendritic branch structure can recover $\Delta Ca_{AP}$. After cutting dendritic branches near the low $\Delta Ca_{AP}$ site, differences in the bAP amplitude, calcium channel conductance, and the

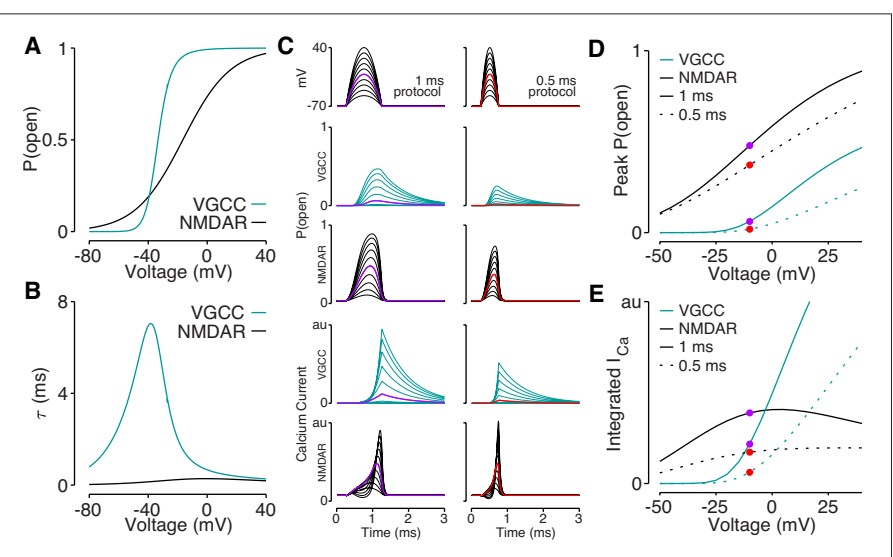

**Figure 8.** Reductions in bAP amplitude have a selective effect on VGCC-mediated calcium influx. (**A**) Steady-state open probability of VGCCs and NMDARs as a function of voltage due to voltage-dependent activation of VGCCs and Mg²⁺-block of NMDARs. (**B**) Voltage-dependent time constant of the activation gate for VGCCs and Mg²⁺ block for NMDARs. (**C**) We simulated the response of VGCCs and NMDARs to a 1ms (left) or 0.5ms (right) quadratic depolarizing voltage with varying amplitude (1st row) designed to resemble bAPs. Voltage-dependent open-probability (2nd and 3rd rows) and voltage-dependent calcium current (4th and 5th rows) for VGCCs (teal) and NMDARs (black). Purple and red traces correspond to summary data in panels D and E. (**D**) Peak open probability of VGCCs and NMDARs during quadratic voltage protocols in panel C. (**E**) Integral of calcium influx through VGCCs and NMDARs during quadratic voltage protocols in panel C.

A-Type potassium channel density required to generate bAPs that peak at –10 mV were all nullified (*Figure 7B–F*). We also computationally recapitulated our results with 4-AP (*Figure 3C–D*) by artificially removing A-Type potassium channels (*Figure 7—figure supplement 2A-D*). These data confirm that dendritic branch structure directly affects local dendritic excitability. Furthermore, these results support the model that increased dendritic branching around low $\Delta Ca_{AP}$ recording sites is sufficient to reduce bAP amplitude and $\Delta Ca_{AP}$ .

Branch structure-dependent changes in dendritic excitability may also affect the amplitude, kinetics, and efficacy of synaptic input arriving at each branch. Using our compartment models, we simulated synaptic input by injecting a conductance into each dendritic site (*Figure 7G–H*). As expected, the local EPSP amplitude was smaller in low $\Delta Ca_{AP}$ branches because of their reduced input resistance (*Figures 7G and 6F*). Although EPSP size measured at the soma (but evoked in the dendrites) was lower for synapses on low $\Delta Ca_{AP}$ branches, the difference was minimized due to frequency-dependent cable attenuation (*Figure 7H*, effect size dendritic EPSPs: –3.23, effect size somatic EPSPs: –1.4). The differences in local EPSP amplitude were abolished after cutting dendritic branches near the low $\Delta Ca_{AP}$ site (*Figure 7—figure supplement 2E-F*). Therefore, although dendritic branch structure affects the local amplitude of synaptic potentials, it has a minimal impact on the efficacy of synaptic input at depolarizing the soma.

## Reductions in bAP amplitude have a selective effect on VGCC-mediated calcium influx

Our data support a model in which elaborated dendritic branching reduces dendritic excitability, which lowers bAP amplitude and bAP-evoked calcium influx. However, it remains to be determined why bAP-dependent amplification of synaptic NMDAR-mediated calcium influx is spared by reductions in bAP amplitude that impact bAP-evoked calcium influx through VGCCs. Using published data (*Jahr and Stevens, 1990*; *Reuveni et al., 1993*), we plotted the voltage-dependent open-probabilities and time constants for VGCCs and NMDARs (*Figure 8A–B*). Using these parameters, we simulated VGCC and NMDAR conductances and the resulting voltage-dependent calcium influx in response to quadratic depolarizing voltage stimuli designed to resemble bAPs (*Figure 8C*). Due to the steep voltage-sensitivity and long time constant of VGCCs (*Figure 8A–B*), they exhibit a dramatic drop-off of calcium influx for voltage steps peaking below –10 mV (*Figure 8C–E*). Although the open-probability of NMDARs is highly sensitive to peak voltage (*Figure 8C–D*), the interaction between open-probability and driving force minimizes differences in calcium influx evoked across a wide range of voltages (*Figure 8E*). These data demonstrate how small reductions in bAP amplitude can eliminate $\Delta Ca_{AP}$ mediated by VGCCs, while largely sparing bAP-dependent amplification of synaptically evoked, NMDAR-mediated calcium influx.

## Discussion

Back-propagating APs regulate synaptic plasticity by evoking voltage-dependent calcium influx throughout dendrites. Here, we show that bAP-dependent calcium influx varies in a dendrite branch-specific manner in cortical L2/3 pyramidal cells due to branch-specific reductions in input resistance inherited from the local morphology of the dendritic tree. Sections of the dendritic tree with more elaborate branch patterns have lower input resistance, leading to a branch-specific reduction in bAP amplitude. These branches have a selective reduction in bAP-evoked calcium influx through VGCCs despite containing VGCCs and successfully propagating bAPs in the absence of synaptic input. However, these branches maintain bAP-dependent amplification of synaptically-evoked calcium influx through NMDARs due to the shallower voltage-dependence and faster kinetics of the $Mg^{2+}$-block of NMDARs. Our results demonstrate that dendritic excitability, bAPs, and bAP-dependent calcium signals vary between branches, which may provide a mechanism for synaptic plasticity rules and the computational properties of dendrites to vary across subcompartments of individual neurons.

### Branch-specific variation in dendritic morphology shapes voltage-dependent calcium signals

Many studies have demonstrated that bAP amplitude and bAP-evoked calcium influx are attenuated in the distal compartments of pyramidal cells (*Regehr et al., 1989*; *Spruston et al., 1995*;

*Magee and Johnston, 1997*; *Waters et al., 2003*; *Froemke et al., 2005*; *Sjöström and Häusser, 2006*; *Larkum et al., 2007*). Our work additionally demonstrates that bAP amplitude and $\Delta Ca_{AP}$ can vary across branches, independent of distance from the soma. This variance can be explained by the branch structure of the dendritic tree. Previous studies have focused on attenuation of $\Delta Ca_{AP}$ as a function of dendritic branch order (*Spruston et al., 1995*; *Williams and Stuart, 2000*); however, we find that explaining the observed variance in $\Delta Ca_{AP}$ requires consideration of the full dendritic morphology, including the number of branches distal to the recording site and on sister dendrites. Elaborate branching patterns reduce input resistance, local bAP amplitude, and bAP-evoked calcium influx because dendritic input resistance is proportional to the surface area of membrane. Although our experiments show that the effect of branch structure on impedance is sufficient to cause the branch-specific differences we observed, additional mechanisms, such as variation in potassium channel density, may also contribute. We note that because bAPs are time varying signals, it is necessary to consider the complex impedance of each dendrite to fully explain bAP attenuation. As an alternative, we use computational models that account for complex impedance to demonstrate that bAP attenuation is predicted by the dendritic branching structure.

Our data cannot precisely resolve how much bAPs are attenuated in low $\Delta Ca_{AP}$ branches because loose-patch recordings do not directly measure intracellular voltage (*Figure 4*) and voltage imaging signals are not precisely calibrated to absolute voltage (*Figure 5*). However, due to the differential impact on VGCC- and NMDAR-mediated calcium influx, we hypothesize that bAPs peak between –40 mV and –15 mV in low $\Delta Ca_{AP}$ branches, based on our analysis in *Figure 8*. Testing this prediction with voltage indicators would require precise calibration of dendritic voltage-dependent fluorescence, which requires knowledge of the dendritic resting potential and a perturbation to a known voltage (such as 0 mV). These results provide a key insight into the mechanisms underlying the intracellular heterogeneity of dendritic biophysical properties and bolster our understanding of dendritic physiology.

### Activation of VGCCs in low $\Delta Ca_{AP}$ branches

Our data indicate that bAP-dependent activation of VGCCs is branch-specific in cortical L2/3 pyramidal cells. Low $\Delta Ca_{AP}$ branches exhibit little to no calcium influx evoked by single bAPs or even bursts of 5 bAPs at 150 Hz. In vivo measurements of L2/3 pyramidal cells have not reported high frequency bursts of bAPs (*Smith et al., 2013*; *Wei et al., 2020*), suggesting that in general, somatic depolarization does not lead to activation of VGCCs in low $\Delta Ca_{AP}$ branches without additional depolarization from local synaptic input. We demonstrated that dendritic glutamate application can evoke VGCC-mediated calcium influx in low $\Delta Ca_{AP}$ dendrites; however, this approach was designed to maximize local depolarization, rather than to resemble a physiological stimulus.

The lack of bAP-dependent activation of VGCCs in low $\Delta Ca_{AP}$ branches may have important implications for neuronal function. First, because VGCC-mediated influx is responsible for multiple forms of synaptic and cellular plasticity (*Kapur et al., 1998*; *Dolmetsch et al., 2001*; *Yasuda et al., 2003*; *Nevian and Sakmann, 2006*; *Scheuss et al., 2006*, *Brigidi et al., 2019*), these data suggest that each of these forms of plasticity may be branch-specific in L2/3 pyramidal cells, at least with respect to their activation by bAPs. Second, these results have clear significance for the interpretation of dendritic calcium signals in vivo. A fundamental challenge in measuring dendritic activity with calcium imaging in vivo is distinguishing between calcium influx evoked by local synaptic input and global calcium signals, such as those evoked by bAPs (*Xu et al., 2012*; *Beaulieu-Laroche et al., 2019*; *Francioni et al., 2019*; *Kerlin et al., 2019*). Our results indicate that in low $\Delta Ca_{AP}$ branches, bAPs do not evoke calcium signals without the addition of local synaptic input, which could make them a useful site for measuring local dendritic processing.

### Implications for synaptic plasticity

Previous work has demonstrated that bAP-evoked calcium influx selectively drives LTD while bAP-dependent amplification of synaptic calcium influx selectively drives LTP (*Feldman, 2000*; *Matsuzaki et al., 2004*; *Nevian and Sakmann, 2006*; *Lee et al., 2009*). Therefore, identifying branches with a selective reduction in bAP-evoked calcium influx while maintaining bAP-dependent amplification of synaptically mediated calcium influx suggests that the ratio of LTD to LTP varies across branches in L2/3 pyramidal cells. Since the ratio of STDP-dependent depression to potentiation determines the

cooperativity, strength, and sparsity of synaptic inputs (*Song et al., 2000*; *Froemke et al., 2005*; *Sjöström and Häusser, 2006*), our work suggests that synaptic tuning properties may vary in a branch-specific manner based on the local dendritic branch structure. These findings complement a series of papers showing that calcium-dependent plasticity rules vary as a function of distance from the soma (*Golding et al., 2002*; *Froemke et al., 2005*; *Sjöström and Häusser, 2006*; *Gordon et al., 2006*).

Our experiments were conducted in juvenile mice (between P21 and P28). Although the dendritic branch structure is static after this age (*Koleske, 2013*; *Richards et al., 2020*), it is possible that changes in channel composition dampen the effects of dendritic branch structure on excitability. We show that such effects are possible with compartment models (*Figure 7F*), and it is known that changes in dendritic excitability can occur as a result of synaptic plasticity induction (*Frick et al., 2003*; *Frick et al., 2004*; *Losonczy et al., 2008*). Therefore, our findings may suggest a developmental function of low $\Delta Ca_{AP}$ branches in shaping synaptic connectivity, consistent with the lower incidence of calcium influx in specific branches of L2/3 pyramidal cells during development in visual cortex (*Yaeger et al., 2019*).

### Implications for dendritic computation

Our work suggests that the dendritic branch structure of low $\Delta Ca_{AP}$ branches can expand the representational capacity of neurons (*Häusser and Mel, 2003*). First, the lower input resistance of low $\Delta Ca_{AP}$ branches makes them more compartmentalized from the rest of the cell, such that they may be more biased toward performing local computations (*Polsky et al., 2004*). Interestingly, our compartment models indicate that compartmentalization is asymmetric (*Williams, 2004*), such that forward propagation of synaptic input from low $\Delta Ca_{AP}$ branches to the soma may be equivalent to high $\Delta Ca_{AP}$ branches, despite their different biophysical properties. Second, branch-specific variation in calcium-dependent plasticity signals may diversify synaptic tuning, which would increase dendritic computational capacity (*Poirazi et al., 2003*; *Häusser and Mel, 2003*; *Francioni and Harnett, 2021*; *Bicknell and Häusser, 2021*). Together, these considerations indicate that low $\Delta Ca_{AP}$ branches may be hotspots for dendritic computation due to their enhanced dependence on cooperative synaptic processing and higher likelihood of containing heterogeneous synaptic tuning.

## Materials and methods
### Genetic constructs

For voltage imaging, we used QuasAr6a, an improved Archaerhodopsin-derived near-infrared genetically encoded voltage indicator (GEVI) (*Wang et al., 2021*). To improve expression and membrane trafficking we created the construct CAG::QuasAr6a-TS-dmCitrine-TSx3-ER2-p2a-jRGECO1a-CAAX, following the design in *Adam et al., 2019*, where TS is the trafficking sequence from Kir2.1 (*Gradinaru et al., 2010*), dmCitrine is the non-fluorescent Y66G mutant of mCitrine (*Adam et al., 2019*), and ER2 is the endoplasmic reticulum export signal FCYENEV (*Gradinaru et al., 2010*). The self-cleaving p2a linker enabled bicistronic co-expression of a membrane-targeted $Ca^{2+}$ indicator, jRGECO1a-CAAX. In the present studies, this indicator was only used for locating expressing neurons; for consistency with other experiments in this manuscript, we used the blue-shifted dye, Fluo-5F for $Ca^{2+}$ imaging.

The genes were cloned into a second-generation lentiviral backbone with a CAG promoter (Addgene: #124775) using standard Gibson Assembly. Briefly, the vector was linearized by double digestion using BamHI and EcoRI (New England Biolabs, Ipswich, MA) and purified by the GeneJET gel extraction kit (ThermoFisher, Waltham, MA). DNA fragments were generated by PCR amplification and then fused with the backbones using NEBuilder HiFi DNA assembly kit (New England Biolabs). All plasmids were verified by sequencing (GeneWiz, Cambridge, MA).

### In utero electroporation (IUE)

All procedures involving animals were in accordance with the National Institutes of Health Guide for the care and use of laboratory animals and were approved by the Harvard University Institutional Animal Care and Use Committee (IACUC, Protocol #: IS00000571-3). The IUE surgery was made as described previously (*Kwon and Sabatini, 2011*). Embryonic day 15.5 (E15.5) timed-pregnant female CD1 mice (Charles River, Wilmington, MA) were deeply anesthetized and maintained with 2% isoflurane. The animal body temperature was maintained at 37 °C. Uterine horns were carefully exposed,

and periodically rinsed with warm PBS. The plasmid DNA was diluted with PBS (2 µg/µL; 0.005% fast green), and 1 µL of the mixture was injected into the left lateral ventricle of pups. Electrical pulses (40 V, 50ms duration) were delivered five times at 1 Hz using a tweezers electroporation electrode (CUY650P5; Nepa Gene, Ichikawa, Japan). Injected embryos were placed back into the abdominal cavity, and the surgical wound was sutured with PGCL25 absorbable sutures (Patterson, Saint Paul, MN).

## Slice preparation

Acute coronal slices were prepared from the somatosensory cortex of young adult, C57Bl/6 j wild-type mice (Jackson Labs, Bar Harbor, ME) between postnatal days P21-P28. Animals were anesthetized via inhalation of isoflurane then immediately decapitated. The brain was rapidly removed and placed in ice-cold cutting artificial cerebrospinal fluid (ACSF) containing (in mM): 125 NaCl, 2.5 KCl, 25 $NaHCO_3$, 1.25 $NaH_2PO_4$, 1 $CaCl_2$, 10 $MgCl_2$, and 25 glucose, saturated with 95% $O_2$ and 5% $CO_2$. We cut coronal slices with a VT1200S Vibratome (Leica, Buffalo Grove, IL) while maintained in the cutting solution. Slices were then incubated at 34 °C for 30 min in a recovery ACSF containing (in mM): 92 NaCl, 28.5 $NaHCO_3$, 2.5 KCl, 1.25 $HaH_2PO_4$, 2 $CaCl_2$, 4 $MgCl_2$, 25 glucose, 20 HEPES, 3 Sodium Pyruvate, and 5 Sodium Ascorbate, saturated with 95% $O_2$ and 5% $CO_2$. Slices were next transferred to a 20 °C room in recovery ACSF until use ( > 30 min after slicing) and not held for longer than 8 hr. Experiments were conducted at 34 °C in recording ACSF containing (in mM): 125 NaCl, 2.5 KCl, 25 $NaHCO_3$, 1.25 $NaH_2PO_4$, 1.5 $CaCl_2$, 1 $MgCl_2$, and 25 glucose, saturated with 95% $O_2$ and 5% $CO_2$. All solutions were prepared with osmolality between 300–310 mOsm/kg, adjusted with either water or glucose. For voltage imaging experiments in *Figure 5*, coronal slices were prepared from CD1 mice between P21-P28. The slicing solution contained (in mM): 210 sucrose, 3 KCl, 26 $NaHCO_3$, 1.25 $NaH_2PO_4$, 5 $MgCl_2$, 10 D-glucose, 3 sodium ascorbate and 0.5 $CaCl_2$, and was saturated with 95% $O_2$ and 5% $CO_2$. The slices were transferred to an incubation chamber containing the recording solution (in mM): 124 NaCl, 3 KCl, 26 $NaHCO_3$, 1.25 $NaH_2PO_4$, 2 $MgCl_2$, 15 D-glucose and 2 $CaCl_2$, and was saturated with 95% $O_2$ and 5% $CO_2$.

## Electrophysiology

Somatic whole-cell recordings were acquired from cortical excitatory L2/3 pyramidal cells using IR-DIC. Patch pipettes (2–4.5 MΩ) were filled with an internal solution containing (in mM): 130 K-Gluconate, 10 KCl, 10 HEPES, 4 MgATP, 0.5 $Na_2GTP$, 10 Phosphocreatine-disodium salt, 0.3 Fluo-5F and 0.01 Alexa 594. For voltage-imaging experiments in *Figure 5*, the internal solution contained (in mM): 8 NaCl, 130 $KMeSO_3$, 10 HEPES, 5 KCl, 0.5 EGTA, 4 Mg-ATP, 0.3 $Na_3$-GTP, 0.3 Fluo-5F, and 0.01 JF549i. The pH was adjusted to 7.3 using KOH and osmolality was adjusted to 285–295 mOsm/kg with water. In a subset of experiments shown in *Figure 2E*, 300 µM Fluo-4 was used instead of Fluo-5F, but amplitudes of calcium signals were comparable, so we merged the data. All recordings were performed with a Multiclamp 700B amplifier (Molecular Devices, San Jose, CA). Pipette capacitance was neutralized prior to break-in and the series resistance was fully balanced for all recordings. Series resistance ranged from 7 to 25 MΩ. We elicited APs by injecting a 1–2ms current of 0.5–3.5 nA. The average resting membrane potential was –79.5 mV with a standard deviation of 6.13 mV (N = 420). Recordings were aborted if the cell's membrane potential exceeded –60 mV, if the dendrites begun filling with calcium (as indicated by the baseline Fluo-5F signal), or if the series resistance became too high for us to reliably evoke action potentials. For experiments in which we performed glutamate uncaging, we also aborted recordings if the input resistance of the cell exceeded 200 MΩ. When we performed glutamate uncaging, we added the following drugs via bath application: 3.75 mM MNI-Glutamate (for uncaging), 10 µM 2-CA (to reduce presynaptic release probability and maintain quiescent recording conditions), 1 unit/mL glutamate pyruvate transaminase (which catalyzes free glutamate into $\alpha$-ketoglutarate), and 3 mM Sodium Pyruvate (a necessary cofactor for GPT), to the recording ACSF. In experiments where we blocked uncaging-evoked depolarization in *Figure 4*, we added 10 µM NBQX to the recording ACSF.

Dendritic loose-patch recordings were acquired from cortical excitatory L2/3 pyramidal cell dendrites after acquiring somatic whole-cell recordings and filling the cells with 10 µM Alexa 594. The open-tip resistance of dendritic pipettes was 8–12 MΩ, and the pipettes were filled with recording ACSF. Dendritic recordings were made with a Multiclamp 700B amplifier (Molecular Devices) in

current-clamp mode. We used scanning DIC and two-photon imaging to target pipettes to dendritic branches. Before acquiring a loose-patch seal, we pushed the dendritic pipette into the dendrite until we observed a visible kink in the dendrite. For each dendritic recording, we recorded the electrical signal evoked by an AP before and after applying a brief pulse of negative pressure to achieve a loose-patch seal (*Figure 4—figure supplement 1*). All dendritic analyses in *Figure 4* are derived from the difference between these two signals, which removes small electrical artifacts of somatic current injection and focuses our analysis on the signal arriving directly from the dendritic membrane.

The glutamate puff solution used in *Figure 3* was applied with a large patch-pipette ( < 1 MΩ) connected to a picospritzer. The puff solution was composed of recording ACSF in addition to 1 mM glutamate, 20 µM CPP and 50 µM MK-801. For glutamate puff experiments, we also added 20 µM CPP and 50 µM MK-801 to the recording ACSF to block NMDARs. In some experiments, we puffed recording ACSF onto the dendrite without glutamate or NMDAR blockers to control for displacement artifacts evoked by the positive pressure (*Figure 3—figure supplement 1*). We used scanning DIC and two-photon imaging to bring puff pipettes near dendrites and used a short 5–15ms puff to apply glutamate to dendritic branches. The pressure of the puff was increased gradually from 5 to 20 PSI until a clear somatic depolarization and dendritic calcium signal were observed.

## Two-photon imaging and glutamate uncaging

Two-photon imaging was performed with a custom-built microscope described previously (*Carter and Sabatini, 2004*). We tuned a mode-locked femtosecond laser to 840 nm to excite both Fluo-5F and Alexa 594. All calcium imaging was performed >10 min after break-in to allow fluorophores to passively diffuse into the dendritic tree. Stimulus-evoked changes in fluorescence were quantified as the change in green (Fluo-5F) fluorescence relative to a baseline period, divided by the baseline red (Alexa 594) fluorescence (ΔG/R) (*Bloodgood and Sabatini, 2007*). We used a custom algorithm to center the field of view on each trial (*Carter and Sabatini, 2004*). We measured fluorescence with a 2ms line-scan through the recording sites (at least 1 dendritic shaft and 0–3 dendritic spines depending on the experiment) and averaged over the spatial coordinates of each compartment to extract a time-varying trace (*Figure 1*). We waited at least 10 s in between trials to minimize photodamage and allow the calcium signal to fully return to baseline. In the experiments in *Figure 6—figure supplement 2*, we performed point-scans on dendritic spines to maximize the sampling rate (recorded at 125 kHz and downsampled to 8 kHz to minimize shot noise).

To perform laser photolysis of MNI-glutamate, we used a short, 500 µs femtosecond laser pulse tuned to 725 nm. For each experiment, we gradually increased laser intensity until either a 0.5 mV EPSP was generated or a clear calcium signal in the dendritic spine was evoked. We moved the uncaging spot around the dendritic spine to maximize the EPSP size as an attempt to uncage gluta-mate directly apposed to the postsynaptic density (*Figure 1B and D*).

## Simultaneous voltage and calcium imaging

Interleaved voltage- and calcium-imaging experiments in *Figure 5* were conducted on modified home-built structured illumination epifluorescence microscope previously reported (*Adam et al., 2019*). Briefly, blue illumination was patterned by a digital micromirror device (DMD) and used for structured illumination calcium imaging via the calcium-sensitive dye Fluo-5F. Yellow illumination was also patterned by a DMD and used for structured illumination mapping of dendritic morphology via a dye JF549i. Red illumination was patterned by a holographic spatial light modulator (SLM) and used for structured illumination voltage imaging.

Laser lines from a blue laser (488 nm, 150 mW, Obis LS) and green laser (561 nm, 150 mW, Obis LS) were combined, sent through an acousto-optic modulator for amplitude control, and expanded onto a DMD (V-7000 VIS, ViALUX) for spatial modulation. The DMD was then re-imaged onto the sample via a 25 x water-immersion objective, NA 1.05, Olympus XLPLN-MP. A red laser (MLL-FN-639, 1 W, CNI lasers) was expanded onto an SLM (P1920-0635-HSP8, Meadowlark, Frederick, CO), combined with the blue and yellow lasers via a polarizing beamsplitter, and re-imaged onto the back aperture of the objective. Fluorescence was collected by the objective and imaged onto a sCMOS camera (Hamamatsu Orca Flash 4.0) with the appropriate emission filter for each indicator. Voltage-imaging recordings were acquired at a 1 kHz frame rate. Calcium imaging was recorded at 20 Hz. Two-photon imaging and reconstruction was performed with a custom-built microscope adapted to be combined

with the 1 P illumination. Maximum intensity projections of z-stacks were used to form images of the dendritic arbor.

## Physiology and imaging analysis

All analysis was performed using custom software written in MATLAB and Python. All plots with error bars show mean +/- standard error. To compute the amplitude and peak time of uEPSPs, we first averaged across trials, subtracted the baseline voltage, and applied a median filter with a 1ms window. Then, we found the maximum voltage between 0 and 25ms after uncaging and defined this as the amplitude and peak time. The amplitude of stimulus-evoked calcium signals (ΔG/R) was computed using a 10ms average of the fluorescence signal centered around the average peak time for each recording site. To compute the amplitude of dendritic electrical signals in *Figure 4*, we first computed the average electrical signal in a 0.2ms window surrounding the peak voltage for each trial. The amplitude of each recording was defined as the average peak voltage from the trials in the top 50th percentile signal-to-noise ratio (peak divided by baseline standard deviation). While this method is unsatisfactory for proper comparisons of amplitude across recordings, we only use these data to determine if somatic APs evoke a measurable electrical signal in the dendrites. For measurements of the ratio of $\Delta Ca_{AP}$ from within segment or across branch point pairs (*Figure 6—figure supplement 1A-C*), we only considered pairs if the proximal site had a $\Delta Ca_{AP} > 0.1$ because the ratio of $\Delta Ca_{AP-dist}/\Delta Ca_{AP-prox}$ is not well-defined if the proximal site failed to exhibit AP-evoked calcium influx. $\Delta Ca_{AP}$ never dropped to ~0 and then recovered in more distal sites along the same branch. Effect sizes in *Figure 7G–H* were computed using Cohen's *d*.

For analysis of epifluorescence voltage- and calcium-imaging data in *Figure 5*, we created a spike-triggered average (STA) movie comprising the average bAP-evoked signal. An initial estimate of the STA waveform for each dendritic branch was calculated by averaging the pixel values in a region of interest comprising the 5% of pixels with greatest mean brightness. Maps of the baseline (F) and spike-dependent ($\Delta F$) signals were then calculated pixel-by-pixel by linear regression of the STA movie against a normalized copy of the initial STA waveform estimate, after accounting for a pixel-dependent offset. To account for spatially variable background, watershed segmentation was applied to the $\Delta F$ map to identify regions with high spike-dependent signal. For the region surrounding pixels with the highest $\Delta F$, a pixel-by-pixel plot of $F$ vs. $\Delta F$ was fit to a line via linear regression. The inverse of the slope was the calculated $\Delta F/F_0$.

## Morphological analysis

After all experiments, we collected two-photon z-stacks of each cell to map the dendritic morphology between the soma and the recording sites. In the neurons used in *Figure 6*, we imaged the entire apical dendritic tree after finishing experiments to reconstruct the cells for morphological analyses and compartmental modeling in NEURON. For every recording site, we measured the on-path distance from the soma using maximum z-stack projections of each cell. The distance for each recording site is a slight underestimate of the true on-path distance from the soma because we ignored changes along the z-axis. All imaging was conducted close to the surface of the slice and the maximal angle between the soma and any recording site was 25°, so errors in measured distance were <10%.

We developed a metric for branch complexity that fully accounts for the extent of nearby dendritic branching, independent of the structure of the dendritic tree. First, we defined the term $\beta$ as the number of dendritic branches at a given on-path distance from a recording site (*Figure 6G*). $\beta$ always starts as 2 (one on each side of the recording site) and increases by 1 after every branch point. This is distinct from Scholl analysis as it measures distance based on the on-path distance rather than absolute distance. Next, we performed point-wise multiplication between $\beta$ and an exponential filter to discount for dendritic branches that are further away from the recording site. Finally, we computed branch complexity as $\Omega = \int_0^\infty \beta\left(x\right) * \exp\left(\frac{-x}{\lambda}\right) dx$ , which is largest at sites that are closest to large numbers of dendritic branches and is inversely proportional to electrotonic input resistance.

The true electrotonic input resistance is analytically determined by the cable length constant, which is not feasible to measure in thin caliber L2/3 pyramidal cell dendrites. As an alternative, we estimated how much membrane is simultaneously depolarized by a bAP as an estimate of how much of the dendritic tree can affect the depolarization at a particular site. We used a rapid point-scan imaging approach that allowed us to measure $\Delta Ca_{AP}$ in dendritic spines at >8 kHz (*Figure 6—figure*

*supplement 2A*). We plotted the latency to peak influx (measured from the derivative of fluorescence) as a function of the distance from the soma and used linear regression to estimate the conduction velocity (*Figure 6—figure supplement 2A-B*). We found that the conduction velocity of bAPs in L2/3 dendrites is 154.9 μm/ms. In the same cells used to measure conduction velocity, we plotted the somatic AP waveforms and measured their full-width half-max (*Figure 6—figure supplement 2C-D*). The somatic AP has a duration of about 0.94ms (*Figure 6—figure supplement 2C-D*). To compute the spatial width of a propagating bAP, we multiplied the conduction velocity by the duration (154.9 μm/ms * 0.94ms = 145.6 μm). This estimate of bAP width led to a choice of length constant that maximized the correlation between branch complexity and input resistance (*Figure 6*, *Figure 6—figure supplement 3*).

## Biophysical compartmental modeling

Simulations used in *Figures 6F and 7*, and *Figure 7—figure supplement 1* were performed in the NEURON simulation environment (*Carnevale and Hines, 2006*) using Python (Version 3.7). We reconstructed all 8 neurons used in *Figure 6* with maximum z-projections of the Alexa 594 fluorescence signal. Each compartmental model was composed of a cylindrical soma with length and diameter equal to 12 μm, a 30 μm axon initial segment (AIS) with a 2 μm diameter connected to the soma, a 300 μm axon with a 1 μm diameter connected to the AIS, and a dendritic tree connected to the soma that with identical branch structure to the 8 neurons used in *Figure 6* and a uniform diameter of 1 μm. Our two-photon imaging data did not permit precise estimation of dendritic diameter but from inspection all dendrites had similar diameters. In some experiments, we performed simulated cuts of the dendritic tree in NEURON. For each cell, we removed branches close to the low $\Delta Ca_{AP}$ recording site until the input resistance of the high and low $\Delta Ca_{AP}$ sites were comparable.

We used standard passive parameters: $C_m$ = 1 μF/cm², Rm = 7000 Ω-cm², Ri = 100 Ω-cm. Active conductances were based on a previous model of L2/3 pyramidal cells that was fit to in vivo somatic and dendritic recordings (*Smith et al., 2013*). Dendritic active conductances were completely uniform, such that all differences across dendrites were determined by the distance from the soma and the dendritic branch structure. Each model contained the following channels with given conductance density in pS/μm²: voltage-gated sodium channels (axon: 170, AIS: 2,550 soma: 85, dendrites: 85), voltage-gated potassium channels (axon: 33, AIS: 100, soma: 100, dendrites: 3), M-Type potassium channels (soma: 2.2, dendrites: 1), calcium-dependent potassium channels (soma: 3, dendrites: 3), high voltage-activated calcium channels (soma: 0.5, dendrites: 0.5), and low voltage-activated calcium channels (soma: 3, dendrites: 1.5). Additionally, we added an A-Type potassium channel (*Migliore et al., 1999*) to the soma and dendrites with a variable conductance density that was set for each cell independently. For experiments in *Figures 6F, 7A–E and G–H*, and *Figure 7—figure supplement 1*, we used a Nelder-Mead simplex algorithm (from the SciPy toolbox in Python) to set the A-Type potassium channel conductance density such that backpropagating APs peaked at –10 mV in the high $\Delta Ca_{AP}$ recording site within each pair (values for each site plotted in left-most column of points in *Figure 7F*). The computed density was used for the soma and entire dendritic tree. We chose to set the AP peak at –10 mV because it is consistent with the AP amplitude measured with whole-cell recordings from L2/3 pyramidal cell dendrites > 100 μm from the soma (*Waters et al., 2003*; *Larkum et al., 2007*; *Smith et al., 2013*). The same values for A-Type potassium channel density were used in experiments with simulated cut dendrites. The values shown in *Figure 7F* use the same algorithm to select A-Type potassium channel density for high and low $\Delta Ca_{AP}$ recording sites independently, for both the intact and cut dendritic branch structure.

## Conductance simulations

We computed the voltage-dependent open probability and time constant of VGCCs and NMDARs using published data (*Jahr and Stevens, 1990*; *Reuveni et al., 1993*). For VGCCs, we used the following voltage-dependent forward and backward rate constants ($\alpha_V$ and $\beta_V$, respectively) of the activation and inactivation gates (in mV):

$$\alpha_{V-activation} = \frac{0.055\left(-27-V\right)}{e^{(-27-V)/3.8}-1} \, ms^{-1}mV^{-1}$$

$$\beta_{V-activation} = 0.94e^{(-75-V)/17} \, ms^{-1}$$

$$\alpha_{V-inactivation} = 0.000457 e^{(-13-V)/50} \; ms^{-1}$$

$$\beta_{V-inactivation} = \frac{0.0065}{e^{(-V-15)/28}+1} \; ms^{-1}$$

To compute open probability and time constant for the activation (denoted '$m$') and inactivation (denoted '$h$') gates of VGCCs, we used the following equations: $P_{open}(V) = \alpha_V/(\alpha_V + \beta_V)$, $\tau(V) = 1/(\alpha_V + \beta_V)$. Because the time constant of the inactivation gate is so slow (the minimum time constant is 170ms between –80 and +40 mV), it acts as a constant, so we did not plot it in *Figure 8B*, although it was accounted for in the simulations in *Figure 8C–E*.

For NMDARs (denoted '$n$'), we calculated the voltage-dependent open-probability and time constant as $P_{open}(V) = k_{off}/(k_{on} + k_{off})$ and $\tau(V) = 1/(k_{on} + k_{off})$ based on the measured on and off rates of the $Mg^{2+}$ block (*Jahr and Stevens, 1990*), with $[Mg^{2+}] = 1$ µM:

$$k_{off} = e^{0.017V+0.96} \; ms^{-1}$$

$$k_{on} = \left[Mg^{2+}\right] e^{-0.045V-6.97} \; ms^{-1}M^{-1}$$

For numerical integration in *Figure 8C*, we used Euler's method to compute changes in the state of each gate with $\Delta t = 0.01 \; ms$. We used the following differential equations:

$$\dot{m} = \alpha_{V-activation} - m\left(\alpha_{V-activation} + \beta_{V-activation}\right)$$

$$\dot{h} = \alpha_{V-inactivation} - h\left(\alpha_{V-inactivation} + \beta_{V-inactivation}\right)$$

$$\dot{n} = k_{off} - n\left(k_{off} + k_{on}\right)$$

The open probability of NMDARs is equal to $n$, and the open probability of VGCCs is equal to $m^2 h$. To convert the open probability into a voltage-dependent calcium current, we used a modified version of the Goldman-Hodgkin-Katz current equation with $[Ca]_{in} = 75 \; nM$ and $[Ca]_{out} = 1.5 \; mM$:

$$I_{Ca} = P_{open}(V) * V \frac{[Ca]_{in} - [Ca]_{out} e^{\frac{-2VF}{RT}}}{1 - e^{\frac{-2VF}{RT}}}$$

We used quadratic voltage depolarizations to mimic the shape of APs. For each stimulus, we used a preset amplitude ($V_{amp}$) and duration ($V_{dur}$). Then, we solved the following equation for $a$: $V_{amp} = \left(a\frac{V_{dur}}{2}\right)^2$, such that the stimulus would have a maximum height of $V_{amp}$ along the domain $-\frac{V_{dur}}{2} < t < \frac{V_{dur}}{2}$. Then, we added the equation: $step(t) = V_{amp} - (at)^2$ to the baseline voltage of –70 mV.

To determine the integrated calcium influx in *Figure 8E*, we integrated the calcium current evoked by the voltage step after subtracting the baseline current (NMDARs are partially open at rest).

## Acknowledgements

We thank all members of the Sabatini Lab for comments and advice, in particular R Hakim, E Sayed, P Capelli, S Melzer, K Reinhold, T Kula, A Granger, and S Kim for suggestions on the manuscript. We also thank J Reggiani, N Hahn, W Regehr, C Harvey, and B Bean for helpful suggestions and critical insights. This work was supported by 5F31NS113353-03 from NINDS to ATL, R37NS046579 from NINDS to BLS, NIH R01 1RF1MH117042-01 to AEC, a Vannevar Bush Faculty Fellowship and a Brain Research Foundation Scientific Innovation Award to AEC, and a grant from the Harvard Brain Initiative to BLS and AEC. JDW is a Merck Awardee of the Life Sciences Research Foundation.

## Additional information

### Competing interests

He Tian: has filed a patent on QuasAr6a (Application #: 63247704). Adam E Cohen: has filed a patent for QuasAr6a (Application #: 63247704). The other authors declare that no competing interests exist.

## Funding

| Funder | Grant reference number | Author |
| --- | --- | --- |
| National Institute of Neurological Disorders and Stroke | F31NS113353 | Andrew T Landau |
| National Institute of Neurological Disorders and Stroke | R37NS046579 | Bernardo L Sabatini |
| National Institute of Mental Health | 1RF1MH117042-01 | Adam E Cohen |
| Defense Advanced Research Projects Agency | Vannevar Bush Faculty Fellowship | Adam E Cohen |
| Brain Research Foundation | Scientific Innovation | Adam E Cohen |
| Harvard Medical School | Harvard Brain Initiative | Adam E Cohen Bernardo L Sabatini |
| Life Sciences Research Foundation | Merck Awardee | J David Wong-Campos |

The funders had no role in study design, data collection and interpretation, or the decision to submit the work for publication.

## Author contributions

Andrew T Landau, Conceptualization, Data curation, Formal analysis, Funding acquisition, Investigation, Methodology, Project administration, Software, Validation, Visualization, Writing - original draft, Writing – review and editing; Pojeong Park, J David Wong-Campos, Investigation, Writing – review and editing; He Tian, Resources, Writing – review and editing; Adam E Cohen, Data curation, Formal analysis, Investigation, Methodology, Resources, Writing – review and editing; Bernardo L Sabatini, Conceptualization, Formal analysis, Funding acquisition, Investigation, Methodology, Project administration, Resources, Software, Supervision, Validation, Visualization, Writing - original draft, Writing – review and editing

## Author ORCIDs

Andrew T Landau http://orcid.org/0000-0001-9105-1636
He Tian http://orcid.org/0000-0003-3282-7275
Adam E Cohen http://orcid.org/0000-0002-8699-2404
Bernardo L Sabatini http://orcid.org/0000-0003-0095-9177

## Ethics

All procedures involving animals were in accordance with the National Institutes of Health Guide for the care and use of laboratory animals and were approved by the Harvard University Institutional Animal Care and Use Committee (IACUC) (Protocol #: IS00000571-3).

## Decision letter and Author response

Decision letter https://doi.org/10.7554/eLife.76993.sa1
Author response https://doi.org/10.7554/eLife.76993.sa2

# Additional files

## Supplementary files
• Transparent reporting form

## Data availability
All data and code is posted on the Harvard Dataverse (https://doi.org/10.7910/DVN/ZHNKGE).

The following dataset was generated:

| Author(s) | Year | Dataset title | Dataset URL | Database and Identifier |
|-----------|------|---------------|-------------|-------------------------|
| Landau AT, Park P, Wong-Campos JD, Tian H, Cohen AE, Sabatini BL | 2022 | Dendritic branch structure compartmentalizes voltage-dependent calcium influx in cortical layer 2/3 pyramidal cells | https://dataverse.harvard.edu/dataset.xhtml?persistentId=doi: | Harvard Dataverse, 10.7910/DVN/ZHNKGE |

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
