## [Editor Report]

Synaptic changes in forebrain neurons typically require the conjunction of dendritic action potentials and synaptic activation. Together these signals cause nonlinear changes in calcium influx that then drive plasticity. The strength of these interactions can vary in complex ways. This study uses state of the art methods to convincingly show how some of these complexities arise from known properties of neuronal dendrites and synaptic NMDA receptors.

---

## [Decision Letter]

**Decision letter after peer review:**

Thank you for submitting your article "Dendritic branch structure compartmentalizes voltage-dependent calcium influx in cortical layer 2/3 pyramidal cells" for consideration by *eLife*. Your article has been reviewed by 3 peer reviewers, including Sacha B Nelson as Reviewing Editor and Reviewer #1, and the evaluation has been overseen by Gary Westbrook as the Senior Editor.

Essential revisions:

The points raised in the reviews below should be easily addressed with textual changes to the paper. The clarifications and answers to questions posed by the reviewers should be addressed. The additional analyses requested by reviewer 2 should either be added to the paper, if feasible, or the reason such callibration is not feasible should be mentioned in the paper. Additional suggested experiments (Rev 3 point 3) may improve the paper, but are not required. The query about the 4-AP results (Rev 2 point 6) could perhaps be addressed with some explanation and a reference or two rather than with additional simulations. The point that there is a fourth potential explanation for the low ∆CaAP dendrites (Rev 3) should be acknowledged and discussed.*Reviewer #1 (Recommendations for the authors):*

Although it would increase the impact to more directly link the differences to differences in plasticity at the same dendrites, this is a difficult experiment and may quite reasonably be left to another study. As pointed out by the other reviewers, important details of the number of observations should be better documented.

*Reviewer #2 (Recommendations for the authors):*

1. According to the authors, this study aims to provide "mechanisms by which ca^2+^ dependent plasticity induction may vary throughout individual neurons" (line 75). In addition, there are repeated references to plasticity throughout the manuscript. However, the uEPSP-bAP pairing experiments (which are closest to actual STDP experiments) reveal comparable amplification levels (Figure 2D). Moreover, this study does not include any direct plasticity experiment nor demonstrates any effect of the difference in bAP-associated ca^2+^ signals on plasticity induction. In my opinion, it is not necessary to connect the findings presented here to plasticity. However, if done, experimental evidence that there is a difference in synaptic plasticity between low and high δ ca^2+^ synapses should be provided.

2. Along the same lines, the uEPSP-bAP pairing experiment seems to be the one closest to an STDP experiment. I could not find any data showing a clear difference during uEPSP-bAP pairings between high and low δ ca^2+^ spines. For example, Figure 2D shows a similar level of amplification. Also, is this difference significant?

3. I understand that low and high δ ca^2+^ spines were defined based on the ca^2+^ transient amplitude. Are the observed amplitudes of bAP-associated spine ca^2+^ signals distributed bimodally? Or how was this amplitude criterium determined? When I look at Figure 2B, the distribution looks more like a continuum.

4. Do all spines connected to the same parent branch exhibit high or low δ ca^2+^? Figure 1—figure supplement 1 shows that spine ca^2+^ signals within 6 micrometers are correlated in this respect. Why 6 micrometers? Dendritic segments between branch points can be longer.

5. The statement "97 spines from 33 cells" (line 128) is the only mention of the number of experimental observations. Please add at least to all the figures containing experimental data the number of included observations.

6. One of the interesting results is the finding that 4-AP nullifies the difference between high and low δ ca^2+^ branches (Figure 3C-D). Does the model, as it is currently, recapitulate this result? Can you please clarify how the K^+^ channel distribution in δ ca^2+^ branches was determined in the model?

7. The NEURON simulation should be able to produce a heat-map showing the amplitude of the bAP-associated ca^2+^ signals overlaid on the morphology of a given neuron. Is my assumption correct that such a graph will mostly show a gradual decrease in ca^2+^ signal amplitude as a function of distance to the soma?

8. line 9: "The synaptic tuning properties of neurons are regulated by voltage-dependent calcium influx, which is typically evoked by back-propagating action potentials (bAPs)." I don't understand this sentence. Doesn't synaptic input primarily determine synaptic tuning?

9. line 68: "We discovered branch-specific reductions in bAP amplitude that are independent of distance from the soma." I don't think it is fair to state that these differences are independent of the distance to the soma (see Figure 2E). Distance to the soma is still an important factor, while branch patterning is another one.

*Reviewer #3 (Recommendations for the authors):*

I have a few specific points that I think could further improve the paper.

1. Throughout the paper, the authors should explicitly state their N (cells / animals) and statistics. They should designate significant effects in their summary graphs. It was sometimes difficult to assess the robustness of the data without knowing these details.

2. The experiments in Figure 5 were the most exciting to me, using a combination of voltage and ca^2+^ imaging. These experiments were done using 1-photon microscopy, so they cannot compare dendrites and spines. Nevertheless, there are still interesting differences in low and high ∆CaAP dendrites, with less propagation in the former. But can the authors calibrate their voltage signals? And can they further quantify their time course? In both cases, I think the paper would be stronger with more rigorous analysis.

3. My only disappointment was that the authors did not further extend these experiments. For example, it would have been very interesting to explicitly look at voltage signals around branch points. It also would be intriguing to see how these signals also depend on dendritic complexity. Lastly, it would be great to compare these signals at different types of dendrites (basal, main apical, apical oblique, apical tuft) within or across cells. While these experiments are not strictly necessary, they would make the paper better.

4. The authors devise the metric of "branch impedance", but its derivation and units are not entirely clear. Instead, it seems more like a measure of dendritic complexity, and I wonder if the authors should simply refer to it as such? Of course, if they were able to really measure impedance, that would be very interesting… but also very difficult.

5. The authors test 3 mechanisms for the low ∆CaAP dendrites (1 = no ca^2+^ channels; 2 = no backpropagating APs; 3 = not reaching threshold to activate ca^2+^ channels). While they ultimately focus on dendritic geometry and impedance, it seems like a fourth mechanism is heightened density K^+^ channels in the low ∆CaAP branches (e.g., Gasaparini, Losonczy, Magee). This should be thoroughly addressed in the Discussion.

---

## [Author Response]

Essential revisions:

Thank you to the reviewers and editors for the generally positive comments and for the helpful suggestions. We have tried to answer all of the questions, added new modeling results to the study, and edited the text.

Several points were raised in common by two or more reviewers and are addressed here. These are:

1. We agree that directly linking our findings to dendrite-specific plasticity experiments is an important next step. However, dendrite-specific synaptic plasticity experiments are difficult, as they require knowing the location of the active synapse. This is traditionally done with glutamate uncaging, which can only be used to study plasticity mechanisms that are expressed postsynaptically (typically by a change in AMPA receptors). Depression protocols induced by glutamate uncaging and bAPs have been very difficult to achieve (perhaps because many forms of depression are expressed presynaptically or by lateral diffusion of AMPA receptors and cannot be detected with glutamate uncaging). We agree with the reviewers that this interesting line of experiments is best left for the next study (and graduate student)…

2. For the experimental data, we added the numbers of ROIs, cells, and mice using the notion n=#ROIs / #Cells / #Mice.

3. Performing a calibration of the voltage imaging data would provide insight into the true amplitude of back-propagating action potentials within dendrites. However, at this point, we cannot absolutely calibrate voltage, which requires knowledge of the dendritic resting potential and a perturbation to a known voltage (such as 0 mV). We explain this in the discussion in lines 570-573 and in the response to reviewer 2. Calibration of signals from voltage indicators remains one of the greatest challenges for their use and may only be fully resolved with the use of ratiometric or lifetime-based indicators.

4. In lines 178-180 of the results, we provide 3 mechanistic explanations for the branch-specific variation in voltage-dependent calcium signals. We chose these 3 mechanisms because they represent general explanations that can be tested directly, although each could be realized in numerous ways. We structured our paper in two levels. First, we determine that bAP amplitude is lower than voltage-gated calcium channel threshold (i.e. Mechanism #3, Figures 3-5), then determine why bAP amplitude is reduced (Figures 6-7). We added a sentence to explain this strategy in lines 183-184.

The points raised in the reviews below should be easily addressed with textual changes to the paper.

We have done as requested and responded to each point below.

The clarifications and answers to questions posed by the reviewers should be addressed.

We addressed all points below.

The additional analyses requested by reviewer 2 should either be added to the paper, if feasible, or the reason such callibration is not feasible should be mentioned in the paper.

See General Point 3.

Additional suggested experiments (Rev 3 point 3) may improve the paper, but are not required.

We are in the process of improving our simultaneous voltage- and calcium-imaging rig and look forward to performing such experiments soon.

The query about the 4-AP results (Rev 2 point 6) could perhaps be addressed with some explanation and a reference or two rather than with additional simulations.

We performed additional simulations, which we believe greatly improve the paper. These simulations are now included in Figure 7 —figure supplement 2. Thank you for the suggestion.

The point that there is a fourth potential explanation for the low ∆CaAP dendrites (Rev 3) should be acknowledged and discussed.

See General Point 4.

Reviewer #1 (Recommendations for the authors):Although it would increase the impact to more directly link the differences to differences in plasticity at the same dendrites, this is a difficult experiment and may quite reasonably be left to another study.

Please see General Point 1.

As pointed out by the other reviewers, important details of the number of observations should be better documented.

Thank you, we addressed these observations and responded to them point by point below.

Reviewer #2 (Recommendations for the authors):1. According to the authors, this study aims to provide "mechanisms by which ca^2+^ dependent plasticity induction may vary throughout individual neurons" (line 75). In addition, there are repeated references to plasticity throughout the manuscript. However, the uEPSP-bAP pairing experiments (which are closest to actual STDP experiments) reveal comparable amplification levels (Figure 2D). Moreover, this study does not include any direct plasticity experiment nor demonstrates any effect of the difference in bAP-associated ca^2+^ signals on plasticity induction. In my opinion, it is not necessary to connect the findings presented here to plasticity. However, if done, experimental evidence that there is a difference in synaptic plasticity between low and high δ ca^2+^ synapses should be provided.

Please see General Point 1.

2. Along the same lines, the uEPSP-bAP pairing experiment seems to be the one closest to an STDP experiment. I could not find any data showing a clear difference during uEPSP-bAP pairings between high and low δ ca^2+^ spines. For example, Figure 2D shows a similar level of amplification. Also, is this difference significant?

Thank you for this point. The uEPSP-bAP pairing experiment is indeed the closest to an STDP protocol. As noted, we did not find a difference in calcium influx evoked by uEPSP-bAP pairings in high and low δ ca^2+^ spines, which we explained by modeling the voltage-dependence of NMDARs to short depolarizations in Figure 8. We use this data to support our finding that the bAP enters the dendrite and spine and can still increase NMDAR-dependent Ca influx in the manner predicted to be necessary for STDP LTP.

In terms of synaptic depression, several lines of evidence indicate that the calcium signal evoked by bAPs is the key variable determining the amount of synapse weakening in STDP protocols. Nevian and Sakmann (2006) showed that the amplitude of calcium influx through VGCCs (and not NMDARs) determines the amount of depression evoked in STDP. Additionally, Feldman (2000) and Anisimova et al., (2022) show that synapses are depressed when their activity is uncorrelated with postsynaptic action potentials, validating the extrapolation of bAP-evoked calcium influx to synaptic depression, independent of synaptic input.

3. I understand that low and high δ ca^2+^ spines were defined based on the ca^2+^ transient amplitude. Are the observed amplitudes of bAP-associated spine ca^2+^ signals distributed bimodally? Or how was this amplitude criterium determined? When I look at Figure 2B, the distribution looks more like a continuum.

Thank you for this question. As you pointed out, the observed amplitudes of bAP-evoked calcium influx fell along a continuum (e.g. Figure 2B). We separated data into high and low populations to visualize the differences at the ends of this continuum, not to suggest that these are representative of distinct modes of the distribution. We apologize that this was not clear and have changed the text to better describe the motivation for grouping the data in lines 138 – 143. We also specify that we only perform statistical analyses on the full distribution of the data as opposed to across these two groups.

4. Do all spines connected to the same parent branch exhibit high or low δ ca^2+^? Figure 1—figure supplement 1 shows that spine ca^2+^ signals within 6 micrometers are correlated in this respect. Why 6 micrometers? Dendritic segments between branch points can be longer.

Thank you for this question. We plotted the correlation between spines location within 6 micrometers of each other because this was within our typical field of view. In post-hoc analysis of our data, we found pairs of spines that we happened to record in between branch points (we did not attempt to maximize the number of these pairs while conducting experiments), some of these data points can be found in Figure 6 —figure supplement 1C. The relative distances between the pairs of spines ranged from 19 – 64 µm. In general, the closer spines are to one another, the more similar their fluorescence. Even though some pairs are in between branch points, the distal spine is closer to an elaboration of the dendritic branch pattern (as portrayed in the low ca^2+^ branch in Figure 6) and has less AP-evoked calcium influx.

5. The statement "97 spines from 33 cells" (line 128) is the only mention of the number of experimental observations. Please add at least to all the figures containing experimental data the number of included observations.

Please see General Point 2.

6. One of the interesting results is the finding that 4-AP nullifies the difference between high and low δ ca^2+^ branches (Figure 3C-D). Does the model, as it is currently, recapitulate this result? Can you please clarify how the K^+^ channel distribution in δ ca^2+^ branches was determined in the model?

This is an excellent question. We set the A-Type potassium channel density for each cell such that bAPs in the high δ ca^2+^ branch peaked at -10 mV, which is consistent with dendritic recordings from L2/3 cells (explained in lines 391-394). For each cell, we used the same density throughout the dendritic tree; this is now made clear by an additional sentence in line 434. The other K^+^ channels in the model used the same parameters as previous NEURON simulations of L2/3 cells.

4-AP blocks several isoforms of voltage-gated potassium channels that don’t map directly onto the physiologically defined channel types available for simulation in NEURON. However, to simulate 4-AP application, we measured the peak voltage and calcium conductance of bAPs after blocking A-Type potassium channels and recapitulated all our experimental results. These simulations are presented in Figure 7 —figure supplement 2 and explained in lines 452-453. We think this improves our manuscript greatly, thank you for the suggestion.

7. The NEURON simulation should be able to produce a heat-map showing the amplitude of the bAP-associated ca^2+^ signals overlaid on the morphology of a given neuron. Is my assumption correct that such a graph will mostly show a gradual decrease in ca^2+^ signal amplitude as a function of distance to the soma?

Thank you for this suggestion, we made the plots for two example cells, it is a good demonstration of the branching dependent decrement in AP amplitude and bAP dependent calcium influx. As you predicted, the graph shows a gradual decrease in voltage and calcium influx, but primarily for sections of the dendritic tree with complex branch patterns. We included this as an additional supplemental figure to Figure 7.

8. line 9: "The synaptic tuning properties of neurons are regulated by voltage-dependent calcium influx, which is typically evoked by back-propagating action potentials (bAPs)." I don't understand this sentence. Doesn't synaptic input primarily determine synaptic tuning?

Thank you bringing this up. We have updated this sentence to clarify our meaning. Synaptic tuning properties are indeed determined by synaptic input, but the strength of synaptic input is regulated by voltage-dependent calcium signals that induce synaptic plasticity.

9. line 68: "We discovered branch-specific reductions in bAP amplitude that are independent of distance from the soma." I don't think it is fair to state that these differences are independent of the distance to the soma (see Figure 2E). Distance to the soma is still an important factor, while branch patterning is another one.

Thank you for this point, this was a challenging thing to explain precisely because we are trying to contextualize our findings within the broader literature while also highlighting the novel aspects of our work. We removed the phrase about distance-independence.

Reviewer #3 (Recommendations for the authors):I have a few specific points that I think could further improve the paper.1. Throughout the paper, the authors should explicitly state their N (cells / animals) and statistics. They should designate significant effects in their summary graphs. It was sometimes difficult to assess the robustness of the data without knowing these details.

Thank you, we have done as you requested. See General Point 2.

2. The experiments in Figure 5 were the most exciting to me, using a combination of voltage and ca^2+^ imaging. These experiments were done using 1-photon microscopy, so they cannot compare dendrites and spines. Nevertheless, there are still interesting differences in low and high ∆CaAP dendrites, with less propagation in the former. But can the authors calibrate their voltage signals? And can they further quantify their time course? In both cases, I think the paper would be stronger with more rigorous analysis.

Thank you for these points, the experiments in Figure 5 were also very exciting to us. Please see General Point 3 for a discussion of calibration.

We can quantify their time course. We found that the full width half max of voltage signals was 6.2 ms +/- 0.9 ms for high δ ca^2+^ branches and 8.0 +/- 4.1 ms for low δ ca^2+^ branches. These findings are now in the text in lines 305-307.

3. My only disappointment was that the authors did not further extend these experiments. For example, it would have been very interesting to explicitly look at voltage signals around branch points. It also would be intriguing to see how these signals also depend on dendritic complexity. Lastly, it would be great to compare these signals at different types of dendrites (basal, main apical, apical oblique, apical tuft) within or across cells. While these experiments are not strictly necessary, they would make the paper better.

Thank you for this comment, we agree that these experiments are interesting and are excited to continue them. Voltage imaging is still challenging, and combined calcium and voltage imaging was challenging on a microscope not designed for doing these types of measurements and whole-cell physiology at the same time. We are in the process of building a new system that will make these questions easier to address in the future.

4. The authors devise the metric of "branch impedance", but its derivation and units are not entirely clear. Instead, it seems more like a measure of dendritic complexity, and I wonder if the authors should simply refer to it as such? Of course, if they were able to really measure impedance, that would be very interesting… but also very difficult.

Thank you for this point. Our modeling results support the idea that this morphological metric is directly related to impedance (e.g. Figure 6 —figure supplement 3). However, it is true that this is an empirical relationship and not an analytical one. Because of this, we changed the name of the metric to “branch complexity”, as you suggested. Note that branch complexity scales inversely with input resistance, so instead of inverted the values as we had done originally, we now just report the value of the integral in the formula described in the methods.

We did not derive this formula directly from the cable equation; however, our approach was motivated by the knowledge that the spatial dropoff of voltage signals at steady state is exponential, so we infer that the spatial dependence on nearby dendritic membrane is also exponential at steady state.

We agree that it would be fascinating to directly measure impedance. The closest we came to doing this was in the dendrite patching experiments in Figure 4, but even then, we could not acquire whole-cell recordings.

5. The authors test 3 mechanisms for the low ∆CaAP dendrites (1 = no ca^2+^ channels; 2 = no backpropagating APs; 3 = not reaching threshold to activate ca^2+^ channels). While they ultimately focus on dendritic geometry and impedance, it seems like a fourth mechanism is heightened density K^+^ channels in the low ∆CaAP branches (e.g., Gasaparini, Losonczy, Magee). This should be thoroughly addressed in the Discussion.

Thank you for this suggestion. Please see General Point 4.